# DISTRIBUTED ASSOCIATIVE MEMORY NETWORK WITH ASSOCIATION REINFORCING LOSS

## ABSTRACT

Despite recent progress in memory augmented neural network research, associative memory networks with a single external memory still show limited performance on complex relational reasoning tasks. The main reason for this problem comes from the lossy representation of relational information stored in a content-based addressing memory and its insufficient associating performance for long temporal sequence data. To address these problems, here we introduce a novel Distributed Associative Memory architecture (DAM) with Association Reinforcing Loss (ARL) function which enhances the relation reasoning performance of memory augmented neural network. In this framework, instead of relying on a single large external memory, we form a set of multiple smaller associative memory blocks and update these sub-memory blocks simultaneously and independently with the content-based addressing mechanism. Based on DAM architecture, we can effectively retrieve complex relational information by integrating diverse representations distributed across multiple sub-memory blocks with an attention mechanism. Moreover, to further enhance the relation modeling performance of memory network, we propose ARL which assists a task's target objective while learning relational information exist in data. ARL enables the memory augmented neural network to reinforce an association between input data and task objective by reproducing stochastically sampled input data from stored memory contents. With this content reproducing task, it enriches the representations with relational information. In experiments, we apply our two main approaches to Differential Neural Computer (DNC), which is one of the representative content-based addressing memory model and achieves state-of-the-art performance on both memorization and relational reasoning tasks.

## 1 INTRODUCTION

The essential part of human intelligence for understanding the story and predicting unobserved facts largely depends on the ability of memorizing the past and reasoning for relational information based on the pieces of memory. In this context, research on artificial intelligence has focused on designing human like associative memory network which can easily store and recall both events and relational information from a part of information.

In neural network research, many approaches generally model sequential data with memory systems, such as Long Short Term Memory (LSTM) (Hochreiter & Schmidhuber, 1997) and memory-augmented neural networks (MANN). Especially, recent approach in MANN constructs an associative memory with a content-based addressing mechanism and stores both input data and its relational information to a single external memory. MANN has already proven to be an essential component on many tasks which need long term context understanding (Weston et al., 2014; Sukhbaatar et al., 2015; Graves et al., 2014; 2016; Gulcehre et al., 2018). Also, compared to recurrent neural networks, it can store more information from sequential input data and correctly recall desired information from memory with a given cue. However, even with its promising performance on a wide range of tasks, MANN still has difficulties in solving complex relational reasoning problems (Weston et al., 2015). Since content-based addressing model implicitly encodes data item and its relational information into one vector representation, they often result in a lossy representation of relational information which is not rich enough for solving relational reasoning tasks. To address such weakness, some researches find relational information by leveraging interaction between memory entities

with attention (Palm et al., 2018; Santoro et al., 2018). Others focus on long sequence memorization performance of memory (Trinh et al., 2018; Le et al., 2019; Munkhdalai et al., 2019). Another attempts apply a self-attention to memory contents and explicitly encode relational information to a separate external memory (Le et al., 2020b). However, all those models need to explicitly find relational information among memory entities with highly computational attention mechanism and have to repeatedly recompute it on every memory update.

In this research, we approach the same problem in a much simpler and efficient way which do not require any explicit relational computation, such as self-attention, among memory entities. We hypothesize that lossy representation of relational information (Le et al., 2020b) in MANN is caused by both a single memory based representation and long-temporal data association performance. Although MANN learns to correlate sequential events across time, its representation is not rich enough to reflect complex relational information existing in input data. Therefore, for the enhanced relation learning, we focus on the richness of representation which implicitly embeds associations existing in input data. For this purpose, we introduce a novel Distributed Associative Memory (DAM) architecture which is inspired by how the biological brain works (Lashley, 1950; Bruce, 2001). In DAM, we replace the single external memory with multiple smaller sub-memory blocks and update those memory blocks simultaneously and independently. The basic operations for each associative memory block are based on the content-based addressing mechanism of MANN, but its parallel memory architecture allows each sub-memory system to evolve over time independently. Therefore, similar to the underlying insight of multi-head attention (Vaswani et al., 2017), our memory model can jointly attend to information from different representation subspaces at different sub-memory blocks and is able to provide a more rich representation of the same common input data. To retrieve rich information for relational reasoning, we apply a soft-attention based interpolation to the diverse representations distributed across multiple memories.

Moreover, to enrich long-term relational information in the memory, we introduce a novel association reinforcing loss (ARL) which fortifies data associations of the memory and generally enhances the memorization capacity of MANN. The ARL forces the memory network to learn to reproduce the number of stochastically sampled input data only based on the stored memory contents. As if, other associated pieces of memory are reminded together whenever a person recalls a certain event in his memory, the data reproducing task enables MANN to have better association and memorization ability for input data. It is designed to reproduce a predefined percentage of input representations in the memory matrix on average and, while optimizing two different tasks at the same time, keep the balance between ARL and target objective loss by dynamically re-weighting each task (Liu & Zhou, 2006; Cui et al., 2019).

By combining the above two approaches, DAM, and ARL, our architecture provides rich representation which can be successfully used for tasks requiring both memorization and relational reasoning. We apply our architecture to Differential Neural Computer(DNC) (Graves et al., 2016), which is one of the representative content-based addressing memory, to construct novel distributed associative memory architecture with ARL. DNC has promising performance on diverse tasks but also known to be poor at complex relational reasoning tasks. In experiments, we show that our architecture greatly enhances both memorization and relation reasoning performance of DNC, and even achieves the state-of-the-art records.

## 2   DIFFERENTIABLE NEURAL COMPUTER

We first briefly summarize DNC architecture which is a baseline model for our approaches. DNC (Graves et al., 2016) is a memory augmented neural network inspired by conventional computer architecture and mainly consists of two parts, a controller and an external memory. When input data are provided to the controller, usually LSTM, it generates a collection of memory operators called as an interface vector $\boldsymbol{\xi}_t$ for accessing an external memory. It consists of several *keys* and *values* for read/write operations and constructed with the controller internal state $\boldsymbol{h}_t$ as $\boldsymbol{\xi}_t = W_\xi \boldsymbol{h}_t$ at each time step $t$. Based on these memory operators, every read/write operation on DNC is performed.

During writing process, DNC finds a writing address, $\boldsymbol{w}_t^w \in [0, 1]^A$, where $A$ is a memory address size, along with write memory operators, e.g. write-in *key*, and built-in functions. Then it updates write-in *values*, $\boldsymbol{v}_t \in \mathbb{R}^L$, in the external memory, $\boldsymbol{M}_{t-1} \in \mathbb{R}^{A \times L}$, along with erasing value, $\boldsymbol{e}_t \in$

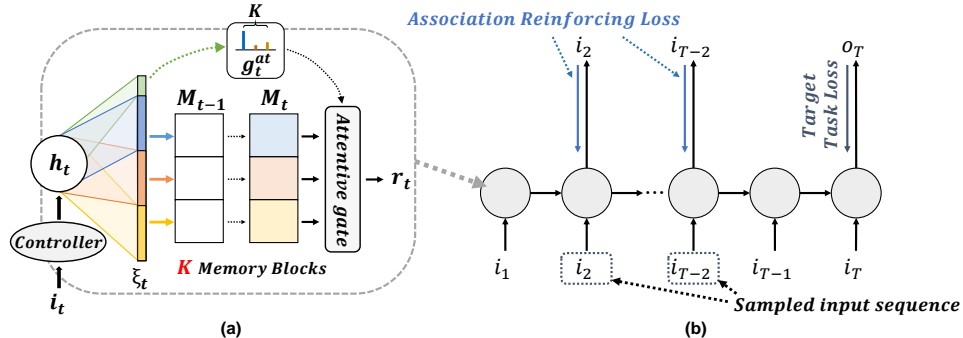

Figure 1: (a) The DAM with $K$ sub-memory blocks and attentive interpolation, $g_t^{at}$. (b) Association Reinforcing Loss.

$[0, 1]^L$, where $L$ is a memory length size as follows:

$$M_t = M_{t-1} \circ (E - w_t^w e_t^\top) + w_t^w v_t^\top \tag{1}$$

where $\circ$ denotes element-wise multiplication and $E$ is $1^{A \times L}$.

In the reading process, DNC searches a reading address, $w_t^{r,i} \in [0, 1]^A$, for $R$ read heads, along with read memory operators, e.g. read-out *key*. Then, it reads out information from the external memory:

$$r_t^i = M_t w_t^{r,i\top} \tag{2}$$

Finally, the output is computed as $y_t = W_y[h_t; r_t] \in \mathbb{R}^{d_o}$, where $r_t = \{r_t^i \in \mathbb{R}^L; 1 \le i \le R\}$. Through these operations, DNC can learn how to store input data and utilize stored information to solve a given task. These whole mechanisms make DNC suitable for a general purposed memory augmented neural network.

## 3 PROPOSED METHOD

In this section, we introduce two methods that improve both memorization and relational reasoning ability of conventional DNC, a distributed associative memory architecture, and an ARL function. For a clear explanation, we illustrate DAM mechanism with a single read head case. For $R$ read head cases of DAM, the details are in the Appendix.

### 3.1 DISTRIBUTED ASSOCIATIVE MEMORY ARCHITECTURE

The distributed associative memory architecture consists of a controller network and $K$ associative memory blocks where each memory block is a content addressable memory similar to the original DNC (Graves et al., 2016). Figure 1(a) shows the overall read/write process of the proposed DAM. For the writing operation, the controller of DAM produces multiple writing operator vectors for multiple memory blocks. Each writing operator vector is used for the content-based addressing of one of the multiple memory blocks, and it is independent of other memory blocks. Since it is produced based on the current input and previous hidden states of the controller, it can independently store its own representation of the same input contents. This writing process enables DAM to store the diverse representations of the same input data to multiple memory blocks with much flexibility. Furthermore, for the reading process, all memory blocks are read at the same time, and read values are interpolated with soft attention to produce single read-out information. Through this attention-based reading process, DAM retrieves the most suitable information for the current task from representations distributed in the multiple memory blocks. Based on these read/write operations, DAM learns how to store and retrieve the diverse representations of input data for different purposed tasks. The following sections detail the main operations.

### 3.1.1 Controller for Multiple Associative Memory Blocks

At each time step $t$, the controller receives an external input, $\boldsymbol{i}_t$, read-out of the previous time step, $\boldsymbol{r}_{t-1}$, and previous hidden state of controller, $\boldsymbol{h}_{t-1}$, to update its current hidden state, $\boldsymbol{h}_t$. After layer normalization, it produces an interface vector, $\boldsymbol{\xi}_t \in \mathbb{R}^{K*(L*R+3L+3R+3)}$, which includes read and write parameters for multiple memory access.

### 3.1.2 Write into Multiple Sub-Memory Blocks

The multiple memory writing processes in our architecture is based on the content-based memory accessing mechanism of DNC. A single memory block is addressed and updated with the same procedure of DNC and such single memory block updating is applied to all blocks independently at the same time. As shown in Eq. (3), each memory block has its own interface vector relevant weight $W_{\xi,1}, \cdots, W_{\xi,k}$, where $k \in \{1, \cdots, K\}$. Theses weights are multiplied with a controller hidden state vector, $\boldsymbol{h}_t$, and used for memory operations of each independent memory block as following.

$$\boldsymbol{\xi}_t = [\boldsymbol{\xi}_{t,1}, \cdots, \boldsymbol{\xi}_{t,K}, \hat{g}_t^{at}] = [W_{\xi,1}, \cdots, W_{\xi,K}, W_{\xi,at}]\boldsymbol{h}_t \tag{3}$$

where $\boldsymbol{\xi}_{t,k}$ is a interface vector for each memory block and $\hat{g}_t^{at}$ is an attentive gate at time $t$.

Based on a writing operator obtained from $\boldsymbol{\xi}_{t,k}$, DAM updates input information into each memory block, $\boldsymbol{M}_{t-1,k}$, independently and simultaneously, following Eq. (1). That independent and simultaneous writing procedures of sub-memory blocks allow that our DAM learns to construct diverse representations for the same common input data.

The following attention-based reading process is designed to integrate representations distributed across sub-memory blocks, and it contributes to enrich representation for relational reasoning tasks.

### 3.1.3 Read from Multiple Sub-Memory Blocks

As in the writing process, DAM obtains a reading operator from $\boldsymbol{\xi}_{t,k}$, and computes reading address, $\boldsymbol{w}_{t,k}^r \in [0,1]^A$, for each memory block. Based on those addresses, DAM reads values from each memory block and derives read-out value, $\boldsymbol{r}_t \in \mathbb{R}^L$, from them, using processed attentive gate, $g_t^{at} \in [0,1]^K$, as follows:

$$\boldsymbol{r}_t = \sum_{k=1}^{K} g_{t,k}^{at} \boldsymbol{M}_{t,k}^{\top} \boldsymbol{w}_{t,k}^r \quad \text{where } g_{t,k}^{at} = Softmax(\hat{g}_{t,k}^{at}) \text{ for } k = 1, \cdots, K. \tag{4}$$

Compared to Eq. (2) of DNC, this reading process integrates representations stored in multiple memory blocks with attention gate and enables DAM to learn to provides the most appropriate distributed representation for a target task.

## 3.2 Association Reinforcing Loss

To enhance the association performance of a memory network, we design a novel auxiliary task, Association Reinforcing Loss (ARL), which can further improve the memorization performance of any given MANN. The main role of ARL task is forcing a memory network to reproduce sampled input data based on its memory content. However, when ARL task is trained with the main target task of the model in a multi-task learning setting, main task related representation and its encoded association can be further emphasized while training (Caruana & De Sa, 1997; Ben-David & Schuller, 2003; Alonso & Plank, 2016; Rei, 2017).

If we define a task-specific target objective function of conventional sequential MANN as follows:

$$\mathcal{L}^{task} = \sum_{t=1}^{T} A(t)\ell_{task}(\boldsymbol{o}_t, \boldsymbol{y}_t) \tag{5}$$

where $T$ is a whole sequence size, $A(t)$ is a function at time $t$, which indicates whether current phase is in answer or not, if its value is 1, then $t$ is in answer phases (otherwise 0). $\boldsymbol{o}_t$ is a target answer and $\ell_{task}(\cdot, \cdot)$ is a task target dependent loss function.

Our ARL function is defined to use a sampled input sequence as its target data as shown in Eq. (6), and this procedure leads the model to rememorize given input information while it is learning the given task.

$$\mathcal{L}_t^{ar} = \ell_{ar}(\boldsymbol{i}_t, \boldsymbol{y}_t) \tag{6}$$

where $\ell_{ar}(\cdot, \cdot)$ is an input sequence dependent loss function., and $\boldsymbol{i}_t$ is an input, $\boldsymbol{y}_t$ is an output at time step $t$, respectively. The error measure for ARL is adopted based on the input item type or main task characteristic. In this research, we use cross-entropy loss or $L2$ loss depending on the given task.

As shown in Fig. 1(b), ARL forces a model to learn to reproduce sampled input sequences from stored representations in memory. When sampling input data, each item of input sequence is sampled with Bernoulli trial with probability, $p$, in which we call as reproducing probability and it is defined as follows:

$$P(\alpha(t) = 1) = 1 - P(\alpha(t) = 0) = p \tag{7}$$

where $\alpha(t)$ is an indicator function that represents sampling status at time $t$.

For an input sequence of length $n$, the series of Bernoulli trial based samplings is same as a Binomial sampling of the input sequence. Therefore, for any input sequence, on average, $np$ samples are reconstructed by ARL because an expected value of Binomial sampling is a product between trial probability, $p$, and the number of trials, $n$. This random sampling policy prevents the model from learning to simply redirect given input to the output of the model at every time step.

When adding ARL to the task-specific target objective for multi-task learning, we also need a new strategy that can control the balance between ARL and original target task loss. Since, as the number of the story input increases, the ARL can overwhelm the total loss of the model. To prevent this loss imbalance problem, we apply a re-weighting method (Cui et al., 2019; Liu & Zhou, 2006), which dynamically keeps the balance between the target task objective $\mathcal{L}^{task}$ and ARL $\mathcal{L}^{ar}$. Moreover, we also introduce a scaling factor, $\gamma$, to ensure the main portion of training loss can be the original target objective function.

$$\gamma = \begin{cases} \hat{\gamma} & \text{if } \hat{\gamma} \geq 1, \\ 1 & \text{otherwise.} \end{cases} \quad \text{where } \hat{\gamma} = \frac{\sum_{t=1}^{T} S(t)\alpha(t)}{\sum_{t=1}^{T} A(t)} \tag{8}$$

where $S(t)$ is an indicator function which represents whether current time step $t$ is in the story phase or not. Finally, the total loss for the training of proposed model follows:

$$\mathcal{L} = \gamma \mathcal{L}^{task} + \sum_{t=1}^{T} \alpha(t)\mathcal{L}_t^{ar} \tag{9}$$

From above two memory related tasks, $\mathcal{L}^{task}$ and $\mathcal{L}^{ar}$, while a model learns to reproduce input representations, target task related representations and its association is further emphasized at the same time. As a result, ARL works as an auxiliary task which reinforces data association for a target objective.

## 4 EXPERIMENTS

We evaluate each of our main contributions, Distributed Associative Memory architecture (DAM) and ARL, separately for ablation study, and show the performance of DAM-AR for complex relational reasoning tasks, such as bAbI, $N^{th}$ farthest task. In all experiments, we adopt well-known neural network generalization techniques that are used in Franke et al. (2018) for our baseline DNC model. The detailed parameter settings and adopted generalization techniques are shown in the Appendix.

### 4.1 DISTRIBUTED ASSOCIATIVE MEMORY ARCHITECTURE EVALUATION

The distributed memory architecture is evaluated in two aspects. First, for model efficiency in data association performance, DAM is configured to have a similar total memory size as a single memory model and evaluated with Representation Recall task. Second, scalability experiments of DAM show the effect of the number of sub-memory blocks on the relation reasoning performance.

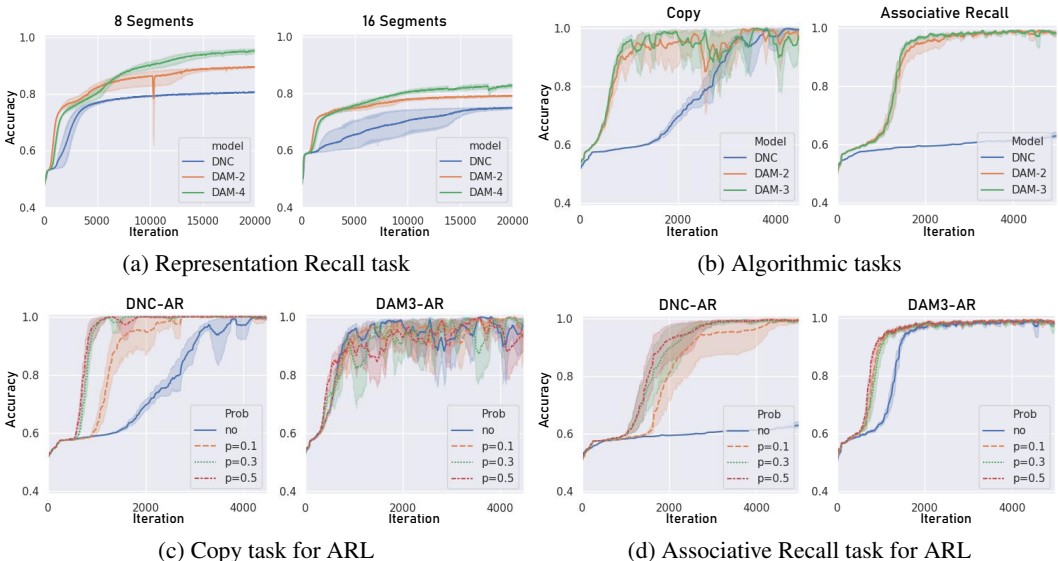

(a) Representation Recall task

(b) Algorithmic tasks

(c) Copy task for ARL

(d) Associative Recall task for ARL

Figure 2: Mean training curves on (a) the Representation Recall task and (b) the algorithmic tasks. Mean training curves for different reproducing probability values on (c) the copy task and (d) the associative recall task. The shadowed area shows a standard deviation of 10 trials.

**Representation Recall Task**   We design a new algorithmic task, called a Representation Recall (RR) task, which evaluates how much representation details a memory model can remember and recall from memory. This task uses randomly generated binary vectors as input sequences. From the sequence, a binary vector is randomly selected and divided into $2N$ sub-parts. Among them, $N$ sub-parts are provided as a cue for a model to predict the rest of sub-parts. In order to solve this task, the model is required to remember $\frac{2N!}{N!(2N-N)!}$ combinations of relations existing in each input, and therefore task complexity increases as $N$ increases. To show the efficiency of a model with a fair comparison, we configure DAM by dividing the original single external memory to $1/2$ or $1/4$ sized sub-memory blocks. The mean training curves of DAM-2 and 4 are compared with original DNC while increasing the task complexity $N$ as in shown Fig. 2(a). The result demonstrates that our proposed architecture learns the task much faster than other DNC based models, and also shows better accuracy and learning stability (smaller standard deviation in learning curve).

**Algorithmic Tasks**   We show the effect of DAM on the copy and the associative recall tasks from Graves et al. (2014). The copy task is designed to show whether a model can store and recall arbitrary long sequential data correctly, and the associative recall task is intended to show whether a model can recall the information associated with a given cue by remembering temporal relation between input data. As shown in Fig. 2(b), simply adopting DAM architecture enhances relation recall performance of the memory model. We can obtain more benefits by adding additional sub-memory blocks (DAM-2 and 3), however, for the copy task, as shown in Fig. 2(c), the effect of the number of memory blocks is small because it is not a task designed for relation reasoning.

**Scalability of DAM**   For the evaluation of the scalability of distributed associative memory architecture without the effect of information loss at a sub-memory block, we adopt a fixed size sub-memory block that has a larger length than a half of the input size and then increase the number of sub-memory blocks to produce several models, DAM-2, 3, and 4. We evaluate all model's performance with complex reasoning tasks, bAbI task, to show the effect of $K$ (representation diversity) on relational reasoning performance. The bAbI task (Weston et al., 2015) is a set of 20 different tasks for evaluating text understanding and reasoning, such as basic induction and deduction. In Fig. 3, the DAM-1 represents a baseline model that has a single external memory and includes modifications from (Franke et al., 2018)

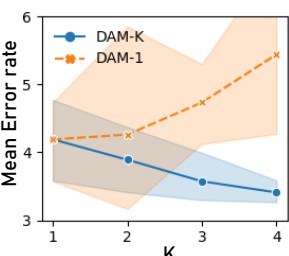

Figure 3: Mean error rate of DAM models

Table 1: Test accuracy [%] on $N^{th}$ farthest task. The results of other models are reported from Santoro et al. (2018); Le et al. (2020b).

| Model | DNC | RMC | TPR | STM | DAM6-AR (0.1) |
|-------|-----|-----|-----|-----|---------------|
| Best  | 25  | 91  | 13  | 98  | 97.8 |

Table 2: (**Top**) The mean word error rate [%] for 10 runs of different DNC based models trained jointly on all 20 bAbI task.(**Bottom**) The mean word error rate [%] for best run of MAMN models trained jointly on all 20 bAbI tasks. The results of other models are reported from Graves et al. (2016); Rae et al. (2016); Franke et al. (2018); Csordás & Schmidhuber (2019); Le et al. (2020a) Dehghani et al. (2018); Munkhdalai et al. (2019); Banino et al. (2020); Le et al. (2020b).

| Model | DNC | SDNC | rsDNC | DNC-MD | NUTM | DAM2-AR (0.1) | DAM2-AR (0.3) |
|-------|-----|------|-------|--------|------|---------------|---------------|
| **Mean** | $16.7 \pm 7.6$ | $6.4 \pm 2.5$ | $6.3 \pm 2.7$ | $9.5 \pm 1.6$ | $5.6 \pm 1.9$ | $\mathbf{1.5} \pm 1.3$ | $2.5 \pm 1.0$ |
| **Best** | 3.8 | 2.9 | 3.6 | n/a | 3.3 | 0.16 | **0.14** |

| Model | Transformer | UT | MNM-p | MEMO | STM | DAM2-AR (0.1) | DAM2-AR (0.3) |
|-------|-------------|-----|-------|------|-----|---------------|---------------|
| **Best** | 22.1 | 0.29 | 0.175 | 0.21 | 0.15 | 0.16 | **0.14** |

for the generalization performance enhancement. For the comparison with DAM-K (K=2, 3, and 4), we linearly increase its single external memory size. The overall graph shows that as the degree of distribution increases, performance on bAbI tasks is also enhanced accordingly for both mean error rate and standard deviation of results. If we use more sub-memory blocks to further increase $K$, we can obtain gradual performance enhancement, which clearly shows the benefits of distributed associative memory architecture.

## 4.2 ASSOCIATION REINFORCING LOSS EVALUATION

To show the effect of ARL on MANN, we apply it to Algorithmic Tasks (Copy and Associative Recall task). In Figs. 2(c) and (d), we show the mean training curves according to the reproducing probability, $p$, on the copy task, and the associative recall task, respectively. For DAM-AR, although we show only DAM3-AR, other configurations(DAM2-AR, DAM4-AR) have similar results. As shown in the figures, the ARL function expedites the learning speed of models in most cases. For the original DNC, it makes model training speed much faster and it is further increased by the high reproducing probability on both tasks. For DAM-AR, the ARL enhances the training speed of the models but DAM is not sensitive to the change of reproducing probability. From those results, we can see that the effect of ARL is related to the property of a given task. For the evaluation of generality of ARL approach, we apply ARL to the other MANN model (RMC) and show the experimental results in Appendix.

## 4.3 DAM-AR EVALUATION ON RELATIONAL REASONING TASKS

As shown in the ablation study, each component of the proposed architecture has a significant impact on the original DNC performance. To show the performance of the whole combined model, DAM-AR, we compare our architecture to other DNC's variations and attention based MANN on following relational reasoning tasks. To support our argument on relational information retrieving approach, we adopts recent memory network models which are applying extensive self-attention (Le et al., 2020b) or multi-head attention for encoding relational information (Santoro et al., 2018) as our counter parts. Specifically, Self-attentive Associative Memory(STM) (Le et al., 2020b) , Relational Memory Core (RMC) (Santoro et al., 2018) and Universal Transformer (UT) (Dehghani et al., 2018) use self-attention and multi-head attention in their memory architecture.

$N^{th}$ **Farthest** This task evaluates a capacity for relational reasoning across time. It asks a model to find the $N^{th}$ farthest vector from a given query vector and this requires the memorization of

Table 3: Test accuracy [%] on Convex hull task. The results of other models are reported from Le et al. (2020b).

| Model | LSTM | ALSTM | DNC | RMC | STM | DAM6-AR (0.3) | DAM8-AR (0.3) |
|---|---|---|---|---|---|---|---|
| $N = 5$ | 89.15 | 89.92 | 89.42 | 93.72 | **96.85** | 95.6 | 95.4 |
| $N = 10$ | 82.24 | 85.22 | 79.47 | 81.23 | **91.88** | 89.8 | 90.5 |

relational information between vectors, such as distance, and sorting mechanism. With this task, long temporal relation modeling performance of a model can be demonstrated.

Table 1 shows a comparison of the $N^{th}$ Farthest task results between our model and other MANN models which are designed for relational reasoning tasks. In the results, even though, the original DNC can not solve the task at all, our DAM-AR shows surprisingly high performance on the task. Even compared to Relational Memory Core (RMC) (Santoro et al., 2018), which explicitly finds relational information from memory based on multi-head attention mechanism, DAM6-AR shows superior performance. For STM (Le et al., 2020b), our model shows a slightly lower accuracy. However, if we consider STM's self-attention computations for finding every possible relations with outer products, our DAM-AR is a quite simple and efficient architecture that does not introduce any explicit relation seeking operations or high-order storage. Therefore, in the aspect of model efficiency, DAM-AR shows comparable performance to STM.

**bAbI QA task** The bAbI task (Weston et al., 2015) is a set of 20 different tasks for evaluating text understanding and reasoning, such as basic induction and deduction. Each task consists of stories for questions and correct answers for the questions, e.g. *Daniel travelled to the bathroom. Mary moved to the office. Where is Daniel? bathroom.* In evaluation, a model is supposed to remember the story and recall related information to provide correct answer for the given questions.

Table 2 shows experimental results on the bAbI task. In this experimental result, our proposed model, DAM2-AR with $p = 0.1$, shows the best mean performance on the bAbI task, among all other DNC based approaches. These results demonstrate that our proposed architecture efficiently learns the bAbI task by using distributed associative memory architecture and association reinforcing loss. Particularly, in Table 2, the best result of DAM2-AR records the state-of-the-art performance on the bAbI task, even compared to other types of recent MANN models.

**Convex hull task** The convex hull task (Vinyals et al., 2015) is predicting a list of points that forms a convex hull sorted by coordinates. The input list is consist of $N$ points with 2D coordinates. In this experiment, we trained the model with $N \in [5, 20]$ and test with $N = 5, 10$ cases. The output is a sequence of 20-dimensional one-hot vectors representing the features of the solution points in the convex-hull. As shown in Table 3, DAM-AR show better accuracy than RMC (Santoro et al., 2018) and similar performance with STM (Le et al., 2020b).

## 5 RELATED WORKS

**Multiple Memory based MANN** In memory slot based MANNs, the content-based addressing is implemented with a dynamic long-term memory which is composed of multiple memory slots (Danihelka et al., 2016; Henaff et al., 2016; Santoro et al., 2018; Goyal et al., 2019). For multiple memory matrix-based models, researchers improve a single memory architecture by adding task-relevant information, asynchronous data input, and relational information to an additional memory matrix (e.g. dual memory) (Munkhdalai & Yu, 2017; Le et al., 2018; 2020b). Our DAM adopts a multiple of the same type of memory matrix for distributed representation. Compared to other approaches, distributed memory architecture is much simpler and shows better performance on the same problems.

**Memory Networks for Relational Reasoning** For relational reasoning, some MANN models explicitly find relational information by comparing their memory entities. RMC (Santoro et al., 2018) leverages interaction mechanisms among memory entities to update memory with relational information. STM (Le et al., 2020b) adopts self-attention for memory contents and store relational

information to separate relation memory. Compared to those methods, DAM provides relational information through diverse representations of input data and long-term association performance of memory.

**Losses for Long-term Dependency**    For long-term memorization of the input pattern, Munkhdalai et al. (2019) used a meta objective loss which forces a model to memorize input patterns in the meta-learning framework. Also, for longer sequence modeling, Trinh et al. (2018) adopted unsupervised auxiliary loss which reconstructs or predicts a sub-sequence of past input data. Compared to Trinh et al. (2018), ARL does not rely on a random anchor point and the sub-sequence reconstruction rather enforces memorization of every past input data that are associated with the target task. ARL focuses on enhancing data association while reproducing input representations, but also considering a balance with target objective loss by applying dynamic weighting method for dual task optimization.

## 6    Conclusion

In this paper, we present a novel DAM architecture and an ARL function to enhance data association performance of memory augmented neural networks. The proposed distributed associative memory architecture stores input contents to the multiple sub-memory blocks with diverse representations and retrieves required information with soft-attention based interpolation over multiple distributed memories. We introduce a novel association reinforcing loss to explicitly improve the long-term data association performance of MANN. Our ARL is designed to reproduce the contents of associative memory with sampled input data and also provides a dynamic task balancing with respect to the target objective loss. We implement our novel architecture with DNC and test its performance with challenging relational reasoning tasks. The evaluation results demonstrate that our DAM-AR correctly stores input information and robustly recall the stored information based on the purpose of the given tasks. Also, it shows that our model not only improves the learning speed of DNC, but reinforce the relation reasoning performance of the model. Eventually, our DAM-AR significantly outperforms all other variations of DNC and shows the state-of-the-art performance on complex relation reasoning tasks, bAbI, even compared to other types of memory augmented network models. As future works, we are going to optimize task balancing strategy between ARL and target loss, and perform further research on how ARL affects multiple sub-memory blocks while optimizing memory model.

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

## A    EXPERIMENT DETAILS

In experiments, for model optimization, we adopt an RMSprop optimizer with a momentum value as $0.9$ and epsilon as $10^{-10}$.

### A.1    MODEL CONFIGURATION DETAILS

We configure hyper-parameters of our model with two types of settings, one for the comparison with other memory network models and the other for the scalability experiment of distributed memory (DM). For the comparison with other models, we keep the almost same total memory size of DNC and divide it into smaller sub-memory blocks to construct a distributed memory system for a fair comparison. Also, we adjust the memory length size of DAM so that it has a similar amount of trainable parameters with other DNC variants. Tables 4 and 5 show our model hyper-parameters for each task, including a controller's internal state, $d_h$, the number of read heads, $R$, the number of memory blocks, $K$, a memory address size, $A$ and a memory length size, $L$.

For the scalability evaluation of DAM, we set the memory length of a single sub-memory block, $L$, same as the memory block size of DAM-2. The size smaller than this causes the information loss because of too small memory matrix size compared to the input length. In this configuration, we increase the degree of distribution by increasing the number of sub-memory blocks. Since we fix the single memory block size, the total memory size increases linearly with $K$. Table 6 shows the details of configuration.

### A.2    EXPERIMENTAL TASK EXPLANATION

In this subsection, we introduce our experimental setup and how to conduct experiments on each task.

#### A.2.1    ALGORITHMIC TASK DESCRIPTION

The experiments on algorithmic tasks are repeated 10 times with a batch size of 16 and a learning rate of $10^{-4}$, and training iterations of 20K on Representation Recall task and training iterations of 10K on Copy and Associative Recall tasks. We evaluate the performance on the algorithmic tasks based on the accuracy which is defined as $L1$ norm between model outputs and targets. In each task, an input flag is provided with story (input) data which are required to produce an answer. After story input, query data along with the output flag are provided to the model. Based on the input and stored information, the model is required to predict the correct answer. For training, we randomly construct each task's data at each iteration as the following configurations.

**Representation Recall Task.**   In this task, $L_i$ binary vectors, each has $W$ length, are randomly generated and are provided to the model along with an input flag. Here, we divide each input binary vector into $2N$ segments and use the half of them, $N$, as a cue vector. For an answer phase, $L_c$ cue vectors are constructed by random sampling from $L_i$ binary vectors with replacement and $N$ segments without replacement. The network has to predict the remaining $N$ segments when each cue vector is provided. We use $L_i = 8$, $W = 64$, $N \in \{2, 4, 8\}$, and $L_c \in [8, 16]$. Therefore, $d_i$ and $d_o$ are 64 and 32, respectively.

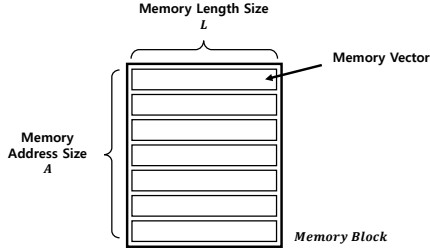

Figure 4: Composition of a memory block.

Table 4: Model hyper-parameters for the copy task, associative recall task, and bAbI task.

| Hyper parameter | Copy | | | Associative Recall | | | bAbI task | |
|---|---|---|---|---|---|---|---|---|
| | DNC | DAM-2 | DAM-3 | DNC | DAM-2 | DAM-3 | DAM-2 | DAM-3 |
| $d_h$ | 128 | 128 | 128 | 128 | 128 | 128 | 256 | 256 |
| $R$ | 1 | 1 | 1 | 1 | 1 | 1 | 4 | 4 |
| $K$ | 1 | 2 | 3 | 1 | 2 | 3 | 2 | 3 |
| $A$ | 64 | 64 | 64 | 32 | 32 | 32 | 128 | 128 |
| $L$ | 36 | 36 | 36 | 36 | 36 | 36 | 48 | 40 |
| Memory Capacity | 2.3K | 4.6K | 6.9K | 1.1K | 2.3K | 3.4K | 12.3K | 15.4K |
| Total Parameters | 0.11M | 0.13M | 0.15M | 0.11M | 0.13M | 0.15M | 0.79M | 0.80M |

Table 5: Model hyper-parameters for the representation recall task, $N^{th}$ farthest task, and the convex hull task.

| Hyper parameter | Representation Recall task | | | $N^{th}$ Farthest task | Convex Hull task | |
|---|---|---|---|---|---|---|
| | DNC | DAM-2 | DAM-4 | DAM-6 | DAM-6 | DAM-8 |
| $d_h$ | 128 | 128 | 128 | 1024 | 256 | 256 |
| $R$ | 1 | 1 | 1 | 4 | 4 | 4 |
| $K$ | 1 | 2 | 4 | 6 | 6 | 8 |
| $A$ | 32 | 32 | 32 | 16 | 20 | 20 |
| $L$ | 120 | 60 | 30 | 128 | 64 | 64 |
| Memory Capacity | 3.8K | 3.8K | 3.8K | 12.3K | 7.7K | 10.2K |
| Total Parameters | 0.23M | 0.20M | 0.18M | 12.6M | 1.5M | 1.7M |

**Copy Task and Associative Recall Task.**    In the copy task (Graves et al., 2014), $L_i$ binary vectors, each has $W$ length, are randomly generated and are provided to the model along with an input flag. After receiving an output flag, the model is required to sequentially produce the same $L_i$ binary vectors as an output. We use $W = 8$ and the number of binary vectors, $L_i \in [8, 32]$, is randomly chosen at each iteration. In the associative task (Graves et al., 2014), we define an item as a sequence of binary vectors with width $W$, and during the input phase, $L_i$ input items, each consists of $N_i$ binary vectors, are provided to a model along with an input flag. Subsequently, a query item, which is randomly chosen from an input item sequence, is provided to the model along with an output flag. In this phase, the model is required to predict a subsequent item that has placed right after the query item in the input item sequence. We use $W = 8$, $N_i = 3$, and the number of items, $L_i \in [2, 8]$, is randomly chosen at each iteration. Therefore, $d_i$ and $d_o$ are 10 on both tasks.

### A.2.2   bAbI Task

Our experiments on bAbI task are repeated 10 times with a batch size of 32 and a learning rate of $[1, \underline{3}, 10] \times 10^{-5}$, and we fine-tuned a model with the learning rate of $10^{-5}$ and training iterations of 10K. We compose experimental set on the bAbI task with 62,493 training samples and 6,267 testing samples and adopt fixed training iterations of about 0.1M. We use a word embedding, which is a general method in a natural language processing domain, with an embedding size of 64. Thus, $d_i$ is 64 and $d_o$ is 160. We evaluate the performance on the bAbI task based on the word error rate for answer words in the test samples.

**Pre-processing on bAbI Task Dataset**    The bAbI task dataset (Weston et al., 2015) consists of many different types of sets that share similar properties. Among them, we only use en-10K set to fairly compare the performance with other researches. In data pre-processing, any input sequences which have more than 800 words are all excluded from the training dataset as in Franke et al. (2018) for computational efficiency. After the exclusion, we construct our dataset with 62,493 training samples and 6,267 testing samples. We remove all numbers from input data and split every sentence into words, and convert every word to its corresponding lower case. After these processes, the

Table 6: Model hyper-parameters for scalability evaluation on the bAbI task.

| Hyper parameter | DAM-1 | DAM-2 | DAM-3 | DAM-4 |
|---|---|---|---|---|
| $d_h$ | 256 | 256 | 256 | 256 |
| $R$ | 4 | 4 | 4 | 4 |
| $K$ | 1 | 2 | 3 | 4 |
| $A$ | 192 | 128 | 128 | 128 |
| $L$ | 64 | 48 | 48 | 48 |
| Memory Capacity | 12.3K | 12.3K | 18.4K | 24.6K |
| Total Parameters | 0.80M | 0.79M | 0.88M | 0.97M |

whole vocabulary composed of 156 unique words and four symbols, which are '[PAD]', '?', '.', and '-'. The '-' symbol represents that the model has to predict an answer word at the current time step. Therefore, answer words are replaced to '-' symbol from input data. The bAbI task dataset is available on `http://www.thespermwhale.com/jaseweston/babi/tasks_1-20_v1-2.tar.gz`.

### A.2.3 $N^{th}$ FARTHEST TASK

We use the same task configuration used in Santoro et al. (2018), which uses Adam Optimizer with a batch size of 1,600 and an initial learning rate of $1e^{-4}$, consists of eight 16-dimensional input vectors, and add 4-layers of MLP with ReLU activation function to the output layer. We set DAM configurations as shown in Table 5.

### A.2.4 CONVEX HULL TASK

We use the same task configuration used in Le et al. (2020b), which adopts RMSProp Optimizer with a batch size of 128 and an initial learning rate of $1e^{-4}$. We add 2-layers of MLP with ReLU activation function (each layer has 256 units) to the output layer and set DAM configurations as shown in Table 5.

## B ADDITIONAL ANALYSIS OF ASSOCIATION REINFORCING LOSS

Proposed Association Reinforcing Loss (ARL) chooses a subset of input sequence based on a stochastic sampling with trial probability, $p$, which is called a reproducing probability. It adaptively decides the number of sampled input story, $n'$, according to the story sequence length, $n$, by adjusting this reproducing probability. Since it is a binomial sampling with independent trials, the expected sampled story length is as shown in Eq. (10). Therefore, it can consistently enhance the memorization performance of memory with the reproducing probability of $p$.

$$E[n'] = np \tag{10}$$

We used an input sequence dependent loss function, $L^{ar}$, of each task as shown in Table 7. We adopt $CrossEntropy$ loss function for the Algorithmic, Representation Recall, and bAbI tasks since their input sequences are the binary vectors or one-hot encoding vectors, whereas $L2$ loss function is adopted to $N^{ar}$ farthest task and the convex hull task because their input data are continuous values.

Table 7: The input data dependent loss function, $L^{ar}$, for each task.

| Task | Algorithmic | Representation Recall | bAbI | $N^{th}$ Farthest | Convex hull |
|---|---|---|---|---|---|
| Loss | | $CrossEntropy$ | | $L2$ | |

## C    MODEL IMPLEMENTATION DETAILS

In this section, we provide more detail of the DAM architecture including well-known neural network generalization techniques that are used in Franke et al. (2018).

At each time step $t$, the controller, LSTM (Hochreiter & Schmidhuber, 1997), receives a input, $\boldsymbol{x}_t \in \mathbb{R}^{d_i}$, previous hidden state, $\boldsymbol{h}_{t-1} \in \mathbb{R}^{d_h}$, and previous memory read-out values, $\boldsymbol{r}_{t-1} = \{\boldsymbol{r}_{t-1}^i \in \mathbb{R}^L; 1 \le i \le R\}$, where $L$ is a memory length size and $R$ is the number of read heads. Based on these values, the controller updates its internal state, $\boldsymbol{h}_t = Controller([\boldsymbol{x}_t; \boldsymbol{r}_{t-1}; \boldsymbol{h}_{t-1}])$. Then, layer normalization (Lei Ba et al., 2016) is applied to the updated internal state. From the normalized internal state $\boldsymbol{h}_{t,LN}$, the controller generates a memory operators, $\boldsymbol{\xi}_t \in \mathbb{R}^{K*(L*R+3L+3R+3)}$, which is called interface vector, as follows:

$$\boldsymbol{\xi}_t = [\boldsymbol{\xi}_{t,1}, ..., \boldsymbol{\xi}_{t,K}, \hat{g}_t^{at}] = [W_{\xi,1}, \cdots, W_{\xi,K}, W_{\xi,at}]\boldsymbol{h}_{t,LN} \tag{11}$$

where $\boldsymbol{\xi}_{t,k} \in \mathbb{R}^{L*R+3L+2R+3}$ is a memory operator for each memory block, $K$ is the number of memory blocks, $k \in \{1, \cdots, K\}$, and $\hat{g}_t^{at} \in \mathbb{R}^{K*R}$ is an attentive gate.

The interface vector, $\boldsymbol{\xi}_{t,k}$, is split into sub-components, and each sub-component and $\hat{g}_t^{at}$ are used for each memory's read/write operations as follows:

$$\boldsymbol{\xi}_{t,k} = [\boldsymbol{k}_{t,k}^w; \hat{\beta}_{t,k}^w; \hat{\boldsymbol{e}}_{t,k}; \boldsymbol{v}_{t,k}; \hat{f}_{t,k}^1, \cdots, \hat{f}_{t,k}^R; \hat{g}_{t,k}^a; \hat{g}_{t,k}^w; \boldsymbol{k}_{t,k}^{r,1}, \cdots, \boldsymbol{k}_{t,k}^{r,R}; \hat{\beta}_{t,k}^{r,1}, \cdots, \hat{\beta}_{t,k}^{r,R}] \tag{12}$$

- the write-in *key* $\boldsymbol{k}_{t,k}^w \in \mathbb{R}^L$;

- the write strength $\beta_{t,k}^w = \zeta(\hat{\beta}_{t,k}^w) \in [1, \infty)$;

- the erase *values* $\boldsymbol{e}_{t,k} = \sigma(\hat{\boldsymbol{e}}_{t,k}) \in [0,1]^L$;

- the write-in *values* $\boldsymbol{v}_{t,k} \in \mathbb{R}^L$;

- R free gates $\{f_{t,k}^i = \sigma(\hat{f}_{t,k}^i) \in [0,1]; 1 \le i \le R\}$;

- the allocation gate $g_{t,k}^a = \sigma(\hat{g}_{t,k}^a) \in [0,1]$;

- the write gate $g_{t,k}^w = \sigma(\hat{g}_{t,k}^w) \in [0,1]$;

- the read-out *keys* $\{\boldsymbol{k}_{t,k}^{r,i} \in \mathbb{R}^L; 1 \le i \le R\}$;

- the read strengths $\{\beta_{t,b}^{k,i} = \zeta(\hat{\beta}_{t,k}^{r,i}) \in [1, \infty); 1 \le i \le R\}$; and

- the attentive gate $\{g_{t,k}^{at,i} = Softmax(\hat{g}_{t,k}^{at,i}) \text{ for } k = 1, \cdots, K.; 1 \le i \le R\}$.

where $\zeta(\cdot)$ denotes a Oneplus function, $\sigma(\cdot)$ denotes a Sigmoid function and $Softmax(\cdot)$ denotes a Softmax function.

Based on those memory operators, the model performs a writing process for each memory block simultaneously. The controller finds a writing address, $\boldsymbol{w}_{t,k}^w \in [0,1]^A$, where $A$ is a memory address size, in two ways: (i) a content-based addressing and (ii) a memory usage statistic.

The content-based addressing computes data address as following:

$$\mathcal{C}(\boldsymbol{M}, \boldsymbol{k}, \beta)[i] = \frac{exp\{\mathcal{D}(\boldsymbol{k}, \boldsymbol{M}[i, \cdot])\beta\}}{\sum_j exp\{\mathcal{D}(\boldsymbol{k}, \boldsymbol{M}[j, \cdot])\beta\}} \tag{13}$$

where $\mathcal{D}(\cdot, \cdot)$ is the cosine similarity, $\boldsymbol{M} \in \mathbb{R}^{A \times L}$ is the memory matrix, and the $\beta$ controls a strength of the address's sharpness.

Therefore, It finds write-content addresses, $\boldsymbol{c}_{t,b}^w \in [0,1]^A$, based on the cosine similarity between write-in *keys* and memory values, as follows:

$$\boldsymbol{c}_{t,k}^w = \mathcal{C}(\boldsymbol{M}_{t,k}, \boldsymbol{k}_{t,k}^w, \beta_{t,k}^w) \tag{14}$$

Next, it finds an allocation address, $\boldsymbol{a}_{t,k} \in [0,1]^A$, considering current memory usage, such as unused or already read memory space. It determines the retention of the most recently read address's

values for each memory block through a memory retention vector, $\boldsymbol{\psi}_{t,k} \in [0,1]^A$ and it calculates current memory usage vector, $\boldsymbol{u}_{t,k} \in [0,1]^A$ as follows:

$$\boldsymbol{\psi}_{t,k} = \prod_{i=1}^{R}(\mathbf{1} - f_{t,k}^i \boldsymbol{w}_{t-1,k}^{r,i}) \tag{15}$$

$$\boldsymbol{u}_{t,k} = (\boldsymbol{u}_{t-1,k} + \boldsymbol{w}_{t-1,k}^w - \boldsymbol{u}_{t-1,k} \circ \boldsymbol{w}_{t-1,k}^w) \circ \boldsymbol{\psi}_{t,k} \tag{16}$$

where $\circ$ denotes a element-wise multiplication.

Afterwards, $\boldsymbol{a}_{t,k}$ is determined based on current memory usage information as follows:

$$\boldsymbol{a}_{t,k}[\phi_{t,k}[j]] = (1 - \boldsymbol{u}_{t,k}[\phi_{t,k}[j]]) \prod_{i=1}^{j-1} \boldsymbol{u}_{t,k}[\phi_{t,k}[i]] \tag{17}$$

where $\phi_{t,k}$ is a free list which informs indices sorted with respect to memory usage values, i.e. $\phi_{t,k}[1]$ is the index of the least used address.

The writing address for each memory block, $\boldsymbol{w}_{t,k}^w$, is determined by interpolation between $\boldsymbol{c}_{t,k}^w$ and $\boldsymbol{a}_{t,k}$, and each memory block is updated, as follows:

$$\boldsymbol{w}_{t,k}^w = g_{t,k}^w[g_{t,k}^a \boldsymbol{a}_{t,k} + (1 - g_{t,k}^a)\boldsymbol{c}_{t,k}^w] \tag{18}$$

$$\boldsymbol{M}_{t,k} = \boldsymbol{M}_{t-1,k} \circ (\boldsymbol{E} - \boldsymbol{w}_{t,k}^w \boldsymbol{e}_{t,k}^\top) + \boldsymbol{w}_{t,k}^w \boldsymbol{v}_{t,k}^\top \tag{19}$$

where $g_{t,k}^a$ controls a proportion between $\boldsymbol{a}_{t,k}$ and $\boldsymbol{c}_{t,k}^w$, $g_{t,k}^w$ controls an intensity of writing, $\circ$ denotes element-wise multiplication and $E$ is $\mathbf{1}^{A \times L}$.

After all memory blocks are updated, the model executes a reading process for each memory block. It finds out a read address for each memory block, $\boldsymbol{w}_{t,k}^{r,i} \in [0,1]^A$, as follows:

$$\boldsymbol{w}_{t,k}^{r,i} = \mathcal{C}(\boldsymbol{M}_{t,k}, \boldsymbol{k}_{t,k}^{r,i}, \beta_{t,k}^{r,i}) \tag{20}$$

where $i \in \{1, ..., R\}$.

Then, it reads preliminary read-out value, $\boldsymbol{r}_{t,k}^i \in \mathbb{R}^L$, from each memory block and interpolates the preliminary read-out values by $g_t^{at}$ to produce read-out value, $\boldsymbol{r}_t^i \in \mathbb{R}^L$, as follows:

$$\boldsymbol{r}_{t,k}^i = \boldsymbol{M}_{t,k}^\top \boldsymbol{w}_{t,k}^{r,i} \tag{21}$$

$$\boldsymbol{r}_t^i = \sum_{k=1}^{K} g_{t,k}^{at,i} \boldsymbol{r}_{t,k}^i \tag{22}$$

where $g_{t,k}^{at,i}$ controls which memory block should be used for final read-out value.

The read-out values, $\boldsymbol{r}^t = \{\boldsymbol{r}_t^i; 1 \le i \le R\}$, are provided to the controller. As in Franke et al. (2018), the model applies drop-out (Srivastava et al., 2014) to $\boldsymbol{h}_{t,LN}$ with a drop-out probability, $p_{dp}$, which is computed as $\boldsymbol{h}_{t,dp} = DropOut(\boldsymbol{h}_{t,LN}|p_{dp})$. Finally, the controller produces an output, $\boldsymbol{y}_t \in \mathbb{R}^{d_o}$, along with $\boldsymbol{h}_{t,dp}$ and $\boldsymbol{r}^t$, as follows:

$$\boldsymbol{y}_t = W_y[\boldsymbol{h}_{t,dp}; \boldsymbol{r}^t] \tag{23}$$

## D   ADDITIONAL RESULTS ON BABI TASK

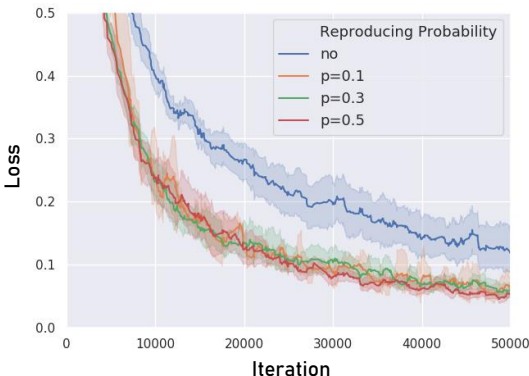

Figure 5: Mean convergence curves of DAM2-AR for different reproducing probability on the bAbI task. The shadowed area shows a standard deviation of 10 trials.

## E   VISUALIZATION OF DISTRIBUTED MEMORY OPERATION (ATTENTIVE GATE)

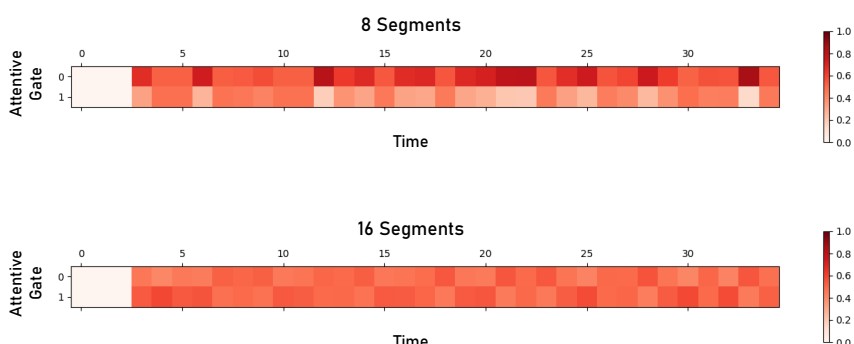

Figure 6: The activated attentive gate of DAM-2 according to time step on Representation Recall task. (**Top**) 8 sub-parts. (**Bottom**) 16 sub-parts.

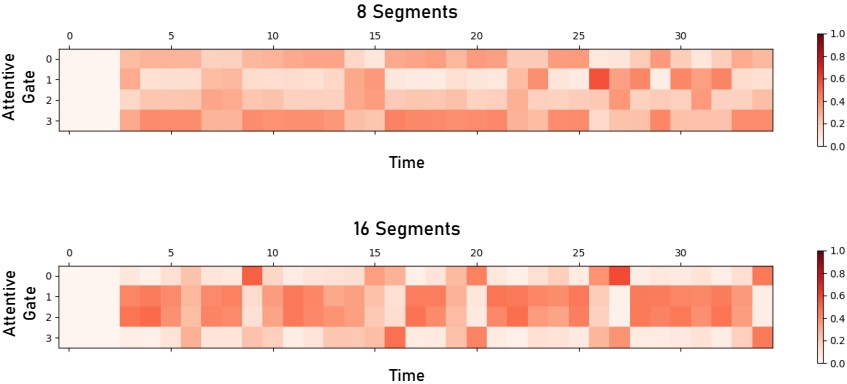

Figure 7: The activated attentive gate of DAM-4 according to time step on Representation Recall task. (**Top**) 8 sub-parts. (**Bottom**) 16 sub-parts.

Fig. 6 and Fig. 7 show relative weights of multiple memory blocks (DAM-2 and 4) obtained from attentive gate while performing Representation Recall (RR) task. During RR task, the model has to predict missing sub-parts when given other parts as a clue. The figures show how much each sub memory block is referenced when the model is predicting sub-parts. As shown in the above figures, for given tasks, the model effectively integrates well-distributed information from multiple memory blocks and utilizing all memory blocks for information retrieval.

## F    EVALUATION RESULT ON GENERAL APPLICABILITY OF DAM-AR APPROACH

Table 8: Test accuracy [%] on $N^{th}$ farthest task.

| Model | Accuracy |
|---|---|
| DNC | 25 |
| RMC | 91 |
| STM | **98** |
| RMC-AR (0.3) | 94 |
| DARMC3-AR (0.3) | 96 |
| DAM6-AR (0.3) | 97.8 |

We applied DAM and ARL to RMC model to verify the general effect on other MANN models. As shown in Table 8, RMC-AR which adopted ARL shows superior relational reasoning performance than original RMC. Furthermore, DARMC3-AR which applied both DAM and ARL to RMC model shows even better performance than RMC-AR. Although RMC itself includes multi-head attention for relation finding, which is redundancy to the DAM, we still can obtain performance enhancement. This result shows the generality of proposed approaches for improving the relational reasoning performance of MANN.

## G    ADDITIONAL EXPERIMENTAL ANALYSIS ON DAM-AR

In the experimental section of our paper, we set up each block size as $\{A = 64, L = 36\}$ for the copy task and $\{A = 32, L = 36\}$ for the associative recall task. To show the model performance with a smaller number of memory slots and memory block length, we show the additional experimental analysis.

### G.1    SCALABILITY EXPERIMENT ON ASSOCIATIVE RECALL TASK

To compare with the experimental setup used in the paper, $A = 32$, we perform a scalability experiment on the associative recall task with $A = 16$ and each of $L$ values of $108$, $54$, and $36$ for each block, and show the performance of DAM according to the number of memory blocks, $K$ in Figure 8. The result shown in Figure 8(a) corresponds to the case when a single memory size of DAM-1 is linearly increased and the same memory size is divided to obtain several numbers of smaller memory blocks of DAM-2 and 3. As shown in the result, the smaller blocks provide more accuracy. The Figure 8(c) shows the case when we adopt a single fixed total memory size for DAM-1, 2, and 3, and each of DAM-2 and 3 represents the number of sub memory blocks under the same condition. Similar to (a), as we divide the memory size to obtain more sub memory blocks, better accuracy is obtained if there is no information loss at a single sub memory block. The Figure 8(b) shows the training curves for the case of (c). It shows DAM architecture can expedite the training speed of the model even with the smaller number of memory slots.

### G.2    EXPERIMENTS WITH SMALLER MEMORY BLOCK ADDRESS SIZE

In our previous Algorithmic experiments, for the copy task, the number of input binary vectors are randomly chosen from $L_i \in [8, 32]$, and the memory address size is set up as $A = 64$. For the

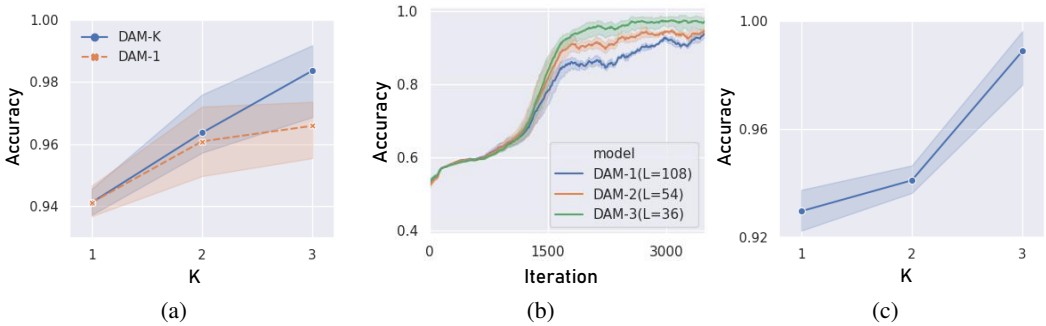

Figure 8: Scalability Experiments with Associative Recall task. (a) Scalability test while increasing single memory size. (b) Training curve while iteratively dividing a fixed memory size. (c) Scalability test while iteratively dividing a fixed memory size.

associative recall task, the number of items is chosen from $L_i \in [2, 8]$ and each input item has $N_i = 3$ length, and $A$ is set to 32. In this experiment, we decreased $A$ from 64 to 16 for the copy task, and from 32 to 16 for the association recall task, so that each memory block has less number of memory slots than the length of the input sequence. Figure 9 shows DAM performance on the copy and the association recall task according to the number of sub memory blocks. In the results, similar to the previous experiments, the memory network performance on both tasks is enhanced as we increase the number of sub memory blocks.

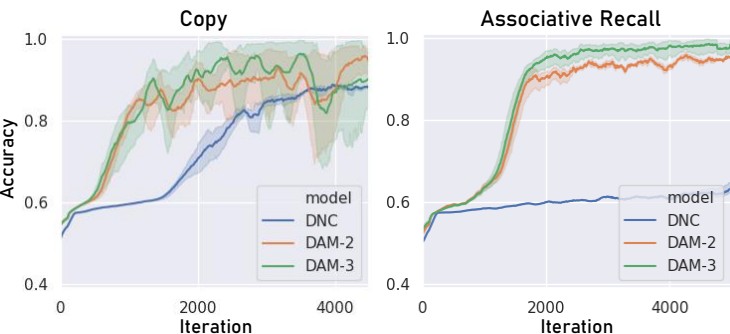

Figure 9: Mean convergence curves on the algorithmic tasks with $A = 16$. The shadowed area shows a standard deviation of 5 trials.

Figure 10 shows the experimental result for the effect of ARL on both copy task and associative recall task. In Figure 10(a), even with the decreased number of memory slots, $A = 16$, ARL still enhances the DNC and DAM performance on the copy task with increasing reproducing probability. For the associative recall task, the result shows a similar pattern as the copy task.

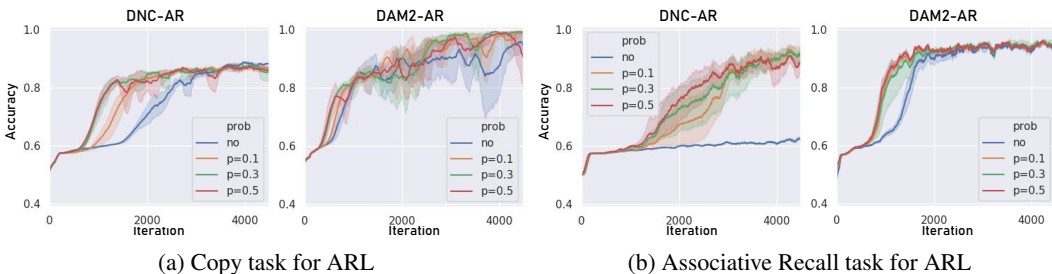

Figure 10: Mean training curves for different reproducing probability values on (a) the copy task with $A = 16$ and (b) the associative recall task with $A = 16$. The shadowed area shows a standard deviation of 10 trials.

## H  DAM MODEL EQUATIONS DIRECTLY RELATED TO DNC

$(DNC)\ \boldsymbol{\xi}_t = W_\xi \boldsymbol{h}_t = [W_{\xi,1}]\boldsymbol{h}_t \in \mathbb{R}^{L*R+3L+5R+3}$

$(DAM)\ \boldsymbol{\xi}_t = W_\xi \boldsymbol{h}_t = [\boldsymbol{\xi}_{t,1}, \cdots, \boldsymbol{\xi}_{t,K}, \hat{g}_t^{at}] = [W_{\xi,1}, \cdots, W_{\xi,K}, W_{\xi,at}]\boldsymbol{h}_t \in \mathbb{R}^{K*(L*R+3L+3R+3)}$

DNC generates the memory operators, $\boldsymbol{\xi}_t$, called as interface vector, for its single memory operation, and DAM extends this vector for multiple independent memory blocks. DAM generates $K$ number of DNC like memory operators, $\boldsymbol{\xi}_{t,k}$, (except for temporal linkage operator) and newly introduce attentive gate, $\hat{g}_t^{at}$ to read from those multiple memory blocks.

$$(DNC)\ \boldsymbol{M}_t = \boldsymbol{M}_{t-1} \circ (\boldsymbol{E} - \boldsymbol{w}_t^w \boldsymbol{e}_t^\top) + \boldsymbol{w}_t^w \boldsymbol{v}_t^\top$$

$$(DAM)\ \boldsymbol{M}_{t,k} = \boldsymbol{M}_{t-1,k} \circ (\boldsymbol{E} - \boldsymbol{w}_{t,k}^w \boldsymbol{e}_{t,k}^\top) + \boldsymbol{w}_{t,k}^w \boldsymbol{v}_{t,k}^\top$$

The writing process of DAM is the same as DNC as shown in the above equations, except the same write operation is executed in multiple memory blocks independently at the same time.

$$(DNC)\ \boldsymbol{r}_t = \boldsymbol{M}_t \boldsymbol{w}_t^{r\top}$$

$$(DAM)\ \boldsymbol{r}_t = \sum_{k=1}^{K} g_{t,k}^{at} \boldsymbol{M}_{t,k}^\top \boldsymbol{w}_{t,k}^r \quad \text{where } g_{t,k}^{at} = Softmax(\hat{g}_{t,k}^{at}) \text{ for } k = 1, \cdots, K.$$

In the reading process of DAM, the basic reading procedure for each memory block is the same as DNC, but, DAM integrates every read-out value from $K$ memory blocks into a single distributed representation with an attentive gate. The attentive gate, $\hat{g}_{t,k}^{at}$, is a newly introduced part of DAM for the attentive interpolation.

Table 9: Results from all runs of our proposed model with DAM2-AR [$p = 0.1$].

| Task | run-1 | run-2 | run-3 | run-4 | run-5 | run-6 | run-7 | run-8 | run-9 | run-10 |
|---|---|---|---|---|---|---|---|---|---|---|
| 1: one supporting fact | 0.0 | 0.0 | 0.0 | 0.0 | 0.0 | 0.0 | 0.0 | 0.0 | 0.0 | 0.0 |
| 2: two supporting facts | 0.1 | 0.6 | 1.6 | 0.0 | 0.3 | 0.5 | 0.4 | 1.1 | 0.2 | 0.5 |
| 3: three supporting facts | 1.1 | 1.1 | 5.1 | 1.2 | 0.8 | 1.2 | 1.5 | 5.6 | 1.1 | 1.2 |
| 4: two argument rels. | 0.0 | 0.0 | 0.0 | 0.0 | 0.0 | 0.0 | 0.0 | 0.1 | 0.0 | 0.0 |
| 5: three argument rels. | 0.5 | 0.5 | 0.4 | 0.8 | 0.4 | 0.5 | 0.6 | 0.7 | 0.2 | 0.5 |
| 6: yes/no questions | 0.0 | 0.1 | 0.0 | 0.1 | 0.0 | 0.0 | 0.1 | 0.0 | 0.0 | 0.0 |
| 7: counting | 1.4 | 1.0 | 2.0 | 1.9 | 0.8 | 0.8 | 1.5 | 1.1 | 0.3 | 1.0 |
| 8: lists/sets | 0.1 | 0.2 | 0.4 | 0.2 | 0.0 | 0.0 | 0.2 | 0.3 | 0.4 | 0.3 |
| 9: simple negation | 0.0 | 0.0 | 0.0 | 0.1 | 0.0 | 0.0 | 0.0 | 0.1 | 0.0 | 0.0 |
| 10: indefinite knowl. | 0.0 | 0.0 | 0.0 | 0.2 | 0.0 | 0.0 | 0.1 | 0.0 | 0.0 | 0.0 |
| 11: basic coreference | 0.0 | 0.0 | 0.0 | 0.0 | 0.0 | 0.0 | 0.0 | 0.0 | 0.0 | 0.0 |
| 12: conjunction | 0.0 | 0.0 | 0.0 | 0.0 | 0.0 | 0.0 | 0.0 | 0.0 | 0.0 | 0.0 |
| 13: compound coref. | 0.0 | 0.1 | 0.0 | 0.0 | 0.0 | 0.0 | 0.0 | 0.2 | 0.0 | 0.0 |
| 14: time reasoning | 0.1 | 0.0 | 0.2 | 0.2 | 0.0 | 0.0 | 0.0 | 0.1 | 0.0 | 0.0 |
| 15: basic deduction | 0.0 | 0.0 | 0.0 | 0.0 | 0.0 | 0.0 | 0.0 | 0.0 | 0.0 | 0.0 |
| 16: basic induction | 0.2 | 0.1 | 51.7 | 49.9 | 0.2 | 0.0 | 45.4 | 42.6 | 51.4 | 0.4 |
| 17: positional reas. | 0.0 | 0.4 | 0.7 | 0.5 | 0.1 | 0.0 | 0.4 | 3.6 | 0.1 | 0.7 |
| 18: size reasoning | 0.0 | 0.2 | 0.0 | 0.2 | 0.2 | 0.2 | 0.3 | 0.6 | 0.1 | 0.1 |
| 19: path finding | 0.2 | 0.4 | 0.7 | 2.6 | 0.3 | 0.3 | 0.6 | 1.5 | 1.6 | 0.0 |
| 20: agent motiv. | 0.0 | 0.0 | 0.0 | 0.0 | 0.0 | 0.0 | 0.0 | 0.0 | 0.0 | 0.0 |
| Mean Err. (%) | 0.19 | 0.24 | 3.14 | 2.90 | 0.16 | 0.18 | 2.56 | 2.88 | 2.77 | 0.24 |

Table 10: Results from all runs of our proposed model with DAM2-AR [$p = 0.3$].

| Task | run-1 | run-2 | run-3 | run-4 | run-5 | run-6 | run-7 | run-8 | run-9 | run-10 |
|---|---|---|---|---|---|---|---|---|---|---|
| 1: one supporting fact | 0.0 | 0.0 | 0.0 | 0.0 | 0.0 | 0.0 | 0.0 | 0.0 | 0.0 | 0.0 |
| 2: two supporting facts | 1.1 | 0.4 | 0.9 | 0.3 | 0.5 | 0.8 | 0.6 | 0.2 | 0.2 | 0.5 |
| 3: three supporting facts | 1.9 | 4.7 | 2.7 | 1.5 | 2.8 | 2.3 | 3.8 | 0.5 | 0.6 | 0.8 |
| 4: two argument rels. | 0.0 | 0.0 | 0.0 | 0.0 | 0.0 | 0.0 | 0.0 | 0.0 | 0.0 | 0.0 |
| 5: three argument rels. | 0.8 | 0.2 | 0.4 | 0.9 | 0.3 | 0.5 | 0.3 | 0.5 | 0.5 | 0.6 |
| 6: yes/no questions | 0.0 | 0.2 | 0.0 | 0.0 | 0.0 | 0.0 | 0.0 | 0.0 | 0.2 | 0.0 |
| 7: counting | 1.5 | 0.7 | 1.3 | 1.7 | 1.2 | 0.3 | 0.8 | 0.4 | 1.6 | 1.4 |
| 8: lists/sets | 0.7 | 0.2 | 0.1 | 0.3 | 1.0 | 0.9 | 0.8 | 0.2 | 0.0 | 0.2 |
| 9: simple negation | 0.0 | 0.0 | 0.0 | 0.0 | 0.0 | 0.0 | 0.0 | 0.0 | 0.0 | 0.0 |
| 10: indefinite knowl. | 0.2 | 0.0 | 0.1 | 0.0 | 0.5 | 0.0 | 0.3 | 0.1 | 0.0 | 0.1 |
| 11: basic coreference | 0.0 | 0.0 | 0.0 | 0.0 | 0.0 | 0.0 | 0.0 | 0.0 | 0.0 | 0.0 |
| 12: conjunction | 0.0 | 0.0 | 0.0 | 0.0 | 0.0 | 0.0 | 0.0 | 0.0 | 0.0 | 0.0 |
| 13: compound coref. | 0.1 | 0.0 | 0.1 | 0.2 | 0.0 | 0.0 | 0.0 | 0.0 | 0.0 | 0.0 |
| 14: time reasoning | 0.0 | 0.0 | 0.1 | 0.0 | 0.0 | 0.1 | 0.0 | 0.3 | 0.2 | 0.0 |
| 15: basic deduction | 0.0 | 0.0 | 0.0 | 0.0 | 0.0 | 0.0 | 0.0 | 0.0 | 0.0 | 0.0 |
| 16: basic induction | 48.7 | 0.4 | 46.5 | 47.2 | 47.2 | 46.0 | 47.6 | 0.1 | 49.3 | 44.6 |
| 17: positional reas. | 0.1 | 0.2 | 1.0 | 1.2 | 0.3 | 0.1 | 18.8 | 0.0 | 0.4 | 0.1 |
| 18: size reasoning | 0.2 | 0.0 | 0.1 | 0.0 | 0.4 | 0.1 | 0.2 | 0.0 | 0.4 | 0.3 |
| 19: path finding | 0.7 | 0.4 | 0.5 | 0.3 | 1.6 | 1.3 | 1.8 | 0.5 | 0.6 | 0.5 |
| 20: agent motiv. | 0.0 | 0.0 | 0.0 | 0.0 | 0.0 | 0.0 | 0.0 | 0.0 | 0.0 | 0.0 |
| Mean Err. (%) | 2.80 | 0.37 | 2.69 | 2.68 | 2.79 | 2.62 | 3.75 | 0.14 | 2.70 | 2.46 |

