# OpenReview forum: "Distributed Associative Memory Network with Association Reinforcing Loss"
_ICLR.cc/2021/Conference — Reject_

### Official Review · AnonReviewer2 · 2020-10-27
**New memory architecture, but unfortunately misses an obvious comparison to self-attention models**

**Rating:** 4
**Confidence:** 4

**Review:**

In this work the authors propose a novel memory architecture wherein memories are stored in multiple ways across a series of memory blocks. By "distributing" the memories in such a manner, the model can flexibly retrieve one version of a memory or another, which enables more flexible computations when conditioning on that memory. The authors demonstrate that such a memory network does well in tasks involving relational reasoning.

One advantage of memory-based models is that they amortize compute across time: they pay an upfront cost to shape and store a memory *online*, but gain an advantages at read time, where they just need to do a basic lookup among the stored memories. A consequence of these  memory-based approaches is that the models need to anticipate how they should store a memory given what might come in the future. In this work, the authors propose the strategy of storing a memory in multiple different ways, which mitigates the risk that a single stored memory will be insufficient for what might come.

In contrast, something like a Transformer pays a heavy cost at read time, as it needs to perform a full self-attention operation (rather than simple lookup) across all stored memories. Transformer-based approaches, however, profit immensely from tasks where memories need to be shaped differently at read time, as they can use the full power of self-attention to morph and condition memories given all the information they've accumulate to that point in time. Given the recent surge of evidence of the usefulness of self-attention based models, and the fact that they can easily be interpreted and/or used as memory models, the authors would be remiss to not include a self-attention based baseline to which they can compare their model. This is especially important given that the complex nature of memory models (i.e., the complexity associated with learning how to read, write, etc.) has recently given way to a more simple approach using memory buffers and self-attention.

This is not to say that there is no value in developing memory-based approaches, as there surely is. However, the memory-based approaches should demonstrate their value in domains that play to their strengths. As stated above, these models amortize the cost of shaping memories over time, and in addition, they can keep a constant size memory indefinitely in time. In contrast, a Transformer-based memory model would grow in memory cost as time increases, and the compute at read time would similarly grow quadratically with time. Thus, to demonstrate the value of a memory-based approach over self-attention, it is wise to pit the two against one another in a regime where self-attention simply becomes too costly; in other words, in a regime where a great number of time steps (and hence memories) need to be considered. Otherwise, the reader is left wondering how well this model compares to the simpler Transformer-based approach.

The authors also propose a new loss that forces the memory contents to be able to predict a sample sequence of previously observed inputs. Unfortunately, I believe the inclusion of this loss makes the absense of a Transformer baseline even more troublesome. This is because, for this loss to be implemented, we need to keep around a buffer of previous inputs, which is precisely the memory cost associated with using a Transformer! So, given the previous discussion on how memory models can in theory maintain a constant sized memory, in the DAM this is no longer the case. Memory costs grow linearly with time because of the need to preserve inputs for use in the ARL loss.

Altogether, the paper is well put together and written well enough to understand the ideas and experiments. The authors did well to choose experiments that would demonstrate the strengths of their approach. Unfortunately, the empirical and rational comparison to Transformer-based approaches prevents me from recommending its publication.

There are a few minor points scattered throughout, but I'll just call attention to the following:

"insufficient associating performance"
--> It is unclear what this means.

"lossy representation"
--> This term is used throughout, but I'm not sure it's warranted. How do we know that the distributed vector representation is truly lossy with respect to the information it must encode? In princple nothing prevents it from being lossless. Given a complicated enough decoder, one wouldn just need a handful of bits to encode very complicated things losslessly.

However, even with its promising performance on a wide range of tasks, MANN still has difficulties in solving complex relational reasoning problems (Weston et al., 2015).
--> There has been much work since 2015 that has improved MANN performance on these tasks. For example, the Sparse DNC, as eventually shown in the results section.

Through this attention-based reading process, DAM retrieves the most suitable information for the current task from distributed representations existing in the multiple memory blocks
--> As shown in this text, and as used throughout, the term "distributed representation" is overloaded in this work. Traditionally the term "distributed representation" is used to denote a vector with real-valued elements, whereas here it is used to denote a set of such vectors, "distributed" across multiple memories.

---

> ### Author Response · Authors · 2020-11-18
> **Response to AnonReviewer2 (1/3)**
>
> **Thank you for your constructive and valuable comments. We’ve revised our paper following the suggestions and will explain your concerns in the following.**
>
> > In contrast, something like a Transformer pays a heavy cost at read time, as it needs to perform a full self-attention operation (rather than simple lookup) across all stored memories. (. . .) This is especially important given that the complex nature of memory models (i.e., the complexity associated with learning how to read, write, etc.) has recently given way to a more simple approach using memory buffers and self-attention.
>
> We already fully compared our model performance with the most recent Transformer based model (UT [3]) and self-attention-based models (RMC [1], STM [2]) to show the effectiveness of our approach. We added more descriptions for other recent MANN models for your understanding. More specifically, RMC adopted multi-head attention to find a relation between its memory slots and UT, Universal Transformer is a generalized Transformer model for complex relational reasoning tasks. STM is ‘self-attentive associative memory’ which uses full self-attention to retrieve relational information from the input sequence. In our experiments, compared to such Transformer-like models, the proposed DAM-AR showed better or comparable relational reasoning performance. Furthermore, as written in the paper, our main contribution is proposing a novel way of relational information retrieving method which can replace the self-attention based approach in Transformer. We repeatedly mentioned relation finding operations of other models in our paper and it mostly corresponds to the self-attention method.
>
>
>
> > This is not to say that there is no value in developing memory-based approaches, as there surely is. (. . .) Otherwise, the reader is left wondering how well this model compares to the simpler Transformer-based approach.
>
> First, recent memory network models we adopted for comparison in our experiments are majorly self-attention based models for relational reasoning. RMC [1] adopts multi-head attention to update the memory contents with its relational information. STM [2], which is an abbreviation of its full name “Self-attentive Associative Memory based Two-memory Model” clearly adopts self-attention for its relation-seeking process based on outer product operation. Furthermore, the model UT [3], Universal Transformer, is an enhanced generalization of Transformer model. Therefore, we think your concerns about ‘not comparing with Transformer-like a model or self-attention models’ can be resolved.
>
> Second, in contrast to the self-attention based models which inevitably introduce quadratic computational overhead for the complex relational reasoning tasks(bAbI, $N^{th}$ farthest), our DAM-AR introduces negligible additional overhead for the same task. Since its basic memory block operation is the same as baseline DNC and the only difference is ‘K’ number of parallel executions of such memory block operations. Furthermore, to obtain these benefits, our model does not need to double or triple the original memory size. Simply dividing its representation length (memory slot length) in two or three subparts was enough modification to achieve similar performance. And DAM-AR’s relationally rich representations even expedite the training speed of the model as shown in Fig.2 of our experiments.
>
> For your concern, to show the overall strength of our approach, we already compared both approaches for the same complex relational reasoning tasks in our experiments. If we consider the computational and memory overhead that self-attention causes, our DAM-AR approach is quite simple and does not introduce additional overhead over the baseline model. Even with such architectural and computational efficiency, our model achieves superior or comparable relational reasoning performance over the self-attention method.
>
>
>
> [1] Santoro, Adam, et al. "Relational recurrent neural networks." Advances in neural information processing systems. 2018.
> [2] Le, Hung, Truyen Tran, and Svetha Venkatesh. "Self-Attentive Associative Memory." arXiv preprint arXiv:2002.03519 (2020).
> [3] Dehghani, Mostafa, et al. "Universal transformers." arXiv preprint arXiv:1807.03819 (2018).

---

> > ### Author Response · Authors · 2020-11-18
> > **Response to AnonReviewer2 (2/3)**
> >
> > > The authors also propose a new loss that forces the memory contents to be able to predict a sample sequence of previously observed inputs. (. . .) Memory costs grow linearly with time because of the need to preserve inputs for use in the ARL loss.
> >
> > Basically, our AR loss does not introduce any buffer for previous sampled input. As mentioned in the paper, it is an additional task loss term simply added to the target objective function without any modification to the network structure. The AR loss is computed at the time of sampling each input data with the Bernoulli trial, therefore we don’t need to store whole sampled input data. Which means, it does not introduce any additional memory space, and memory cost does not grow linearly with time or sequence length because of ARL. It is only adopted during the training phase of the memory model, and it even expedites the learning speed of the given model. Also, its memory cost is much cheaper than that of Transformer since Transformer needs storage for each and every self-attention operation and there are multiple of such layers. For computational complexity, Transformer needs quadratic comparison operations for full self-attention, however, ARL only involves sampling and loss computation for only a sub-portion of the input sequence. Therefore, in terms of both memory usage and computation complexity, AR loss uses far fewer resources than Transformer.
> >
> > If we explain more details, AR loss term enforces the memory network to reproduce sampled input data only based on the stored representation in its memory matrix. And the reproducing input data is stochastically sampled with the Bernoulli trial for each input data (with probability ‘p’). Because AR loss is computed at the time of sampling and added to the target loss, there is no need for buffering or memorizing the store sampled input sequence. Because the Bernoulli trial of each input data consists of Binomial sampling, the average proportion of the sampled input sequence is ‘np’, where n is the length of the input sequence. In other words, AR loss is trying to refresh the stored contents of memory with the percentage of ‘np’. This stochastic sampling scheme of AR loss prevents our model learns to simply redirect input to the output to satisfy AR loss condition, and finally enhances overall memorization performance.
> >
> >
> >
> >
> > > Altogether, the paper is well put together and written well enough to understand the ideas and experiments. The authors did well to choose experiments that would demonstrate the strengths of their approach. Unfortunately, the empirical and rational comparison to Transformer-based approaches prevents me from recommending its publication.
> >
> > As we mentioned above, we compared our model with other most recent state of the art memory network models which adopted Transformer-like approaches (self-attention, multi-head attention). And our model shows superior or comparable performance compared to such relation seeking oriented memory network models.
> >
> > In our paper, one of our main argument is, complex relational information exists in input sequence can be retrieved and stored for further use without the help of extensive relation searching operations, such as full self-attention in Transformer. Therefore, Transformer-like models are our counterpart for comparison, not baseline. In this paper, we are proposing a novel approach for relational information retrieving, which does not heavily rely on self-attention operations. And successfully showed, even without extensive relation searching with self-attention, we can obtain a good memory network model which has the comparable performance of the state-of-the-art MANN. Also, to show that our two main contributions are generally applicable to any MANN, we added the experimental results of our another MANN model (RMC-AR, DMRMC-AR) which integrates our modifications, to the Appendix of our manuscript.
> >
> >
> >
> >
> > > "insufficient associating performance" --> It is unclear what this means.
> >
> > We used the term ‘associating performance’ to represent how much the given memory network represents relational information that exists in input sequences according to a given task. The basic definition of ‘associative’ of associative memory network is “the ability to learn and remember the relationship between unrelated items”. In the same context, we used the term ‘associating performance’ for the ability of the memory network which retrieves the relationship among many items in the input sequences. In other words, how much complex relational information can be encoded to the memory for the following tasks.

---

> > > ### Author Response · Authors · 2020-11-18
> > > **Response to AnonReviewer2 (3/3)**
> > >
> > > > "lossy representation" --> This term is used throughout, but I'm not sure it's warranted. How do we know that the distributed vector representation is truly lossy with respect to the information it must encode? In principle, nothing prevents it from being lossless. Given a complicated enough decoder, one wouldn’t just need a handful of bits to encode very complicated things losslessly.
> > >
> > > The lossy representation term comes from the STM [2] (ICML2020) paper, and it does not mean that representation itself should be perfect to contain every possible information from the input sequence. We used it in the context of relational reasoning performance of stored representation. To avoid confusion, we changed it to “lossy representation of relational information”. The LSTM or content-based addressing network encodes information from an input sequence to a single encoded vector. When trying to solve complex relational reasoning tasks based on sequential input data, usually such representation fails to include complex relations, such as ‘multi-hop’[4], that exist in the input sequence. Therefore, many researchers try to encode relational information in several formats. End-to-End memory network used 2 representations for each hop and others applied a self-attention mechanism to produce separate relational representation (RMC [1], STM, UT [3]). In this context, lossy representation means that a single representation itself does not include rich enough relational information for complex relational reasoning tasks.
> > >
> > >
> > >
> > >
> > >
> > > > However, even with its promising performance on a wide range of tasks, MANN still has difficulties in solving complex relational reasoning problems (Weston et al., 2015). --> There has been much work since 2015 that has improved MANN performance on these tasks. For example, the Sparse DNC, as eventually shown in the results section.
> > >
> > > As shown in our experimental section, we not only included the variants of DNC but also show the result of other state of the art MANN models (RMC [1], UT [3], STM [2], MNM-p [5], MEMO [6]) which aimed to tackle relational reasoning performance with attention mechanism or relying on the different type of method such as meta-learning.
> > >
> > >
> > >
> > >
> > > > Through this attention-based reading process, DAM retrieves the most suitable information for the current task from distributed representations existing in the multiple memory blocks --> As shown in this text, and as used throughout, the term "distributed representation" is overloaded in this work. Traditionally the term "distributed representation" is used to denote a vector with real-valued elements, whereas here it is used to denote a set of such vectors, "distributed" across multiple memories.
> > >
> > > As you pointed out, the way referring the architecturally distributed memory with stored representations could mislead the reader to have incorrect perception about the term ‘distributed representation’ usage. Therefore, we updated the manuscript accordingly to remove confusion. In our paper, considering the traditional concept of distributed representation, we used the term distributed representation, because the main idea is coherent. The concept of distributed representation does not solely mean just denoting a vector with real values. It is more conceptually adopted based on how the vector is constructed. For example, the paragraph vector from PV-DM [7] is obtained by merging several different words that are sampled from the same paragraph to include the semantics of the paragraph. This vector is referred to as a distributed representation of that paragraph. In our case, several different encoded vectors of the same input item (word), each comes from a different representation subspace, are merged to a single vector with an attention mechanism to include more rich relational information for the following task. Therefore, the term “distributed representation” is used based on its conceptual similarity.
> > >
> > >
> > >
> > > [4] Sukhbaatar, Sainbayar, Jason Weston, and Rob Fergus. "End-to-end memory networks." Advances in neural information processing systems. 2015.
> > > [5] Munkhdalai, Tsendsuren, et al. "Metalearned neural memory." Advances in Neural Information Processing Systems. 2019.
> > > [6] Banino, Andrea, et al. "Memo: A deep network for flexible combination of episodic memories." arXiv preprint arXiv:2001.10913 (2020).
> > > [7] Le, Quoc, and Tomas Mikolov. "Distributed representations of sentences and documents." International conference on machine learning. 2014.

---

> > > ### Comment · AnonReviewer2 · 2020-11-18
> > > **Clarification**
> > >
> > > If the AR loss needs to keep track of which inputs have been chosen so far (so that it can sample from them), then it does need to keep a buffer of some sort. I was incorrect to assume that it needed a buffer of the literal inputs, since it could instead keep indices that can be used to find these samples in the data source. Nonetheless, I don't believe it's true that there is no additional memory demand, but I now appreciate that it is negligible.

---

> > > > ### Author Response · Authors · 2020-11-21
> > > > **Response to AnonReviewer2 within (2/3)**
> > > >
> > > > **Thank you for your fast response.**
> > > >
> > > > However, AR loss literally does not need any buffer for sampled inputs. As we explained more specifically, in RNN like sequential processing models, at each time step ‘t’, a single item of the input sequence is provided to the model and **the model also produces output at that time step ‘t’ based on the accumulated information in hidden state**. For DNC it corresponds to the external memory contents. The back-propagation for model training starts to happen at the time step when a model receives a question word as input and target loss is computed. Therefore, we can compute AR loss at each time step and accumulate it before the target loss computation. This is one of the benefits of the Bernoulli trial based sampling scheme. For each time step, we toss a coin to decide whether to sample the current input or not. This is, so to speak, the Bernoulli trial. According to this coin toss, if the current input item should be sampled, then we can **immediately compute AR loss for current input from the predicted output of the model at ‘that time step’**. The error measure (L2 or cross-entropy) is decided by task property or data type. After AR loss computation for the current input item, **it can be stored in a single variable which accumulates all AR loss for another sampled inputs until target loss computation and back-propagation occur**. The only storage we need to keep for AR loss is a single accumulating variable. And this variable can be reset right after the current input sequence ends. Maybe you might expect to use small buffers for implementation convenience, however, the algorithm itself does not require and nothing to do with maintaining buffers. **Furthermore, AR loss is only applied while a model is in training. Therefore, it has no effect on test time**. I hope this very basic explanation helps your understanding of AR loss operations and why it does not need buffers.

---

> > ### Comment · AnonReviewer2 · 2020-11-18
> > **Thanks**
> >
> > Thanks for the response.
> >
> > There is a single comparison to a universal transformer: a citation to previous results on bAbI. What about the other tasks? Also, I'm not sure that UT's are "the most recent Transformer based model", or even the most notable, but to be a bit more explicit in my question: why is the proposed model not compared to a conventional Transformer on all the tasks presented?

---

> > > ### Author Response · Authors · 2020-11-21
> > > **Response to AnonReviewer2 within (1/3)**
> > >
> > > **Thank you for your fast response.**
> > >
> > >
> > > First of all, our main argument is that our approaches, Distributed representation, and ARL, can be applied as simple and efficient **relational information retrieving method** for any MANN, instead of **‘self-attention’** or other attention methods which exhaustively search relational information with quadratic operations. Therefore, our focus is on the method itself for searching relational information, and Transformer is not the state of the art model which use self-attention for relational reasoning. In the experimental section, we already compared our model with **self-attention based state-of-the art performance model, STM**, for all complex relational reasoning tasks. As you mentioned in your review title as your concern, **the self-attention based memory network model is majorly adopted for comparison in our paper. As far as we know, at the time of writing this paper, STM (ICML2020) is the most recent state-of-the-art performance memory network model which uses self-attention for relational reasoning tasks.**
> > > Also, as you mentioned like ‘Transformer **like**’ models, the value of Transformer does lie in its major mechanisms, such as self-attention and multi-head attention. Simply copy and pasting the original Transformer in a specific domain does not always produce good results. Therefore, recent MANNs are adopting the self-attention or multi-head attention concept to their model in their own way. Different from what you expected, conventional Transformer does not have very good performance as a memory network model. Especially, **it is not performing well on smaller or more structured language understanding tasks** and has weakness in complex relational reasoning which is well explained in UT[3]. In other words, conventional Transformer is far from the state-of-the-art performance on relation reasoning tasks. We also updated bAbI task result of Transformer for your reference in **Table 2**. Because of those weaknesses, **Universal Transformer, UT, is proposed by Google** who originally invented Transformer. In Universal Transformer paper, with extensive experiments, they show that **UT is superior to an original Transformer in every aspect when it comes to the tasks which require relational information memory.** Therefore, we already compared our model with a more advanced **direct** extension of Transformer model, and **‘STM showed even better performance than such UT and STM is mainly adopting self-attention for relational reasoning tasks’.**
> > >
> > >
> > > You can reference UT paper for the full comparison between Transformer and UT. Moreover, as far as we know, at the time of writing our paper, UT is the most recent **“directly extended” Transformer** model **“for relational reasoning performance as MANN (for tasks like bAbI, or structured language understanding)"**. Other models are adopting the mechanisms of Transformer or focused on longer input sequence processing. If you know another example of a direct Transformer based memory network model **for the same purposed tasks**, please let us know, it will be very grateful as constructive comments and also helpful for the clarification of our work. The purpose of our experimental comparison is not the showing the comparison result of every pre-existing ‘Transformer like models’ to our model (which is not possible), rather showing the result of **the most related and representative memory network** model with the state-of-the-art performance on relational reasoning, **to support our main argument.**
> > > As far as we know, STM is the most recent (ICML2020) and the best performing model on relational reasoning tasks. It is adopting a full self-attention mechanism, as you wanted, and outperforming all other memory network models for relational reasoning. And it is compared with our model for complex relational reasoning tasks in our paper. Which **strongly supports our argument that our distributed representation based approach is comparable to the self-attention mechanism for memory network**. And as you pointed out in your review, your main focus is also about comparing with self-attention based model which has the state-of-the-art performance.

---

> ### Author Response · Authors · 2020-11-23
> **Have additional concerns?**
>
> Dear AnonReviewer 2,
>
> We believe that we have addressed your concerns and clarified some of your points and revised our paper accordingly. Could you please let us know if you have any additional concerns or questions? We would be happy to provide further revisions or experiments to address any remaining issues and would appreciate a response from you for an updated impression.

---

### Official Review · AnonReviewer3 · 2020-10-29
**Official Blind Review**

**Rating:** 8
**Confidence:** 4

**Review:**

This paper proposes an extension of the Differenciable Neural Computer networks (DNC)
In these DNCs, the reading operation on the external memory are done by accessing a single memory block, which represents a single piece of information or knowledge. The architecture proposed in this paper aims instead to give the possibility to access multiple memory blocks at the same time. In this way, the approach of reading memory is more holistic (Chalmers, 1992). This is a desirable feature of distributed representations and, then, distributed memories. Otherwise, these memories are just similar to the classical approach of representing symbols (Fodor and Pylyshyn ,1988). This debate of what is the main charateristic of distributed representations is revitalized in Ferrone and Zanzotto (2020). It is then a needed extension of DNC.
The paper is well written and results are convincing.
However, there is a minor problem. There is not a direct link among equations in Section 2 and equations in Section 3. Clearly, DNC equations are extended by equations in Section 3. Are these equations linked only with the M, that is the Memory?

References

Fodor, J. A., and Pylyshyn, Z. W. (1988). Connectionism and cognitive architecture: a critical analysis. Cognition 28, 3–71.
Chalmers, D. J. (1992). Syntactic Transformations on Distributed Representations. Dordrecht: Springer.
Ferrone, Zanzotto (2020), Symbolic, Distributed, and Distributional Representations for Natural Language Processing in the Era of Deep Learning: A Survey

---

> ### Author Response · Authors · 2020-11-18
> **Response to AnonReviewer3**
>
> **Thank you for the great feedback. We appreciate your comments for improving the clarity of our manuscript.**
>
> We updated the manuscript accordingly with a further description of DNC components and its related extended equations. Also included which equations are directly related and which component is modified by DAM approach. The main modifications are applied to the interface vector generation part of DNC and attentive gate.
>
> More specifically,
>
>
> (DNC) $ \\xi_t = W_\\xi h_t =  [ W_{ \\xi, 1} ] h_t \\in \\mathbb{R}^{L*R+3L+5R+3} $
>
> (DAM) $ \xi_t= W_\xi h_t = [ \xi_{t,1}, \cdots, \xi_{t,K}, \hat{g}_t^{at} ] $
>
> $ = [ W_{\xi,1}, \cdots, W_{\xi,K}, W_{\xi,at} ] h_t \in \mathbb{R}^{K*(L*R+3L+3R+3)} $
>
> DNC generates the memory operators, $\xi_t$, called as interface vector, for its single memory operation, and DAM extends this vector for multiple independent memory blocks. DAM generates $K$ number of DNC like memory operators, $\xi_{t,k}$, (except for temporal linkage operator) and newly introduce attentive gate, $\hat{g}_{t}^{at}$ to read from those multiple memory blocks.
>
>
> (DNC) $M_t=M_{t-1}\circ(E-w_t^w e_t^\top)+w_t^w v_t^\top$
>
> (DAM) $M_{t,k} = M_{t-1,k} \circ (E-w_{t,k}^w e_{t,k}^\top) + w_{t,k}^w v_{t,k}^\top$
>
> The writing process of DAM is the same as DNC as shown in the above equations, except the same write operation is executed in multiple memory blocks independently at the same time.
>
>
> (DNC) $ r_t = M_t{w_t^r}^\top $
>
> (DAM) $ r_t = \sum_{k=1}^K g_{t,k}^{at} M_{t,k}^\top{w_{t,k}^{r}} $   where   $ g_{t,k}^{at} = Softmax(\hat{g}_{t,k}^{at}) $   for  $ k=1,\cdots,K. $
>
>
> In the reading process of DAM, the basic reading procedure for each memory block is the same as DNC, but, DAM integrates every read-out value from $K$ memory blocks into a single distributed representation with an attentive gate. The attentive gate, $\hat{g}_{t,k}^{at}$, is a newly introduced part of DAM for the attentive interpolation.
>
> Furthermore, we appreciate recommending helpful reference researches of distributed representation.

---

### Official Review · AnonReviewer4 · 2020-10-29
**Convincing demonstration that distributing memory improves learning**

**Rating:** 6
**Confidence:** 4

**Review:**

The authors propose a distributed memory architecture which shares some interface with the Differentiable Neural Computer however crucially segments memory into a collection of K units. The authors show that by increasing K the model learns to use its memory for algorithmic tasks such as copying and associative recall and learn faster. The authors also propose an auxiliary loss to improve memory representations, which involves reconstructing inputs from the representations in memory.

I think the scientific statement is quite clear here and the paper is worth accepting; the only shame is that the authors did not apply this approach to a richer task than bAbI.

Also it would have been nice to compare the approach to a multi-head attention transformer since these also use distributed representations (across heads).

The authors may be interested in the following architecture MERLIN which also uses a reconstruction loss to improve memory representations: https://arxiv.org/abs/1803.10760

---

> ### Author Response · Authors · 2020-11-18
> **Response to AnonReviewer4**
>
> **Thank you for your constructive and valuable comments. We’ve revised our paper following the suggestions and will explain your concerns in the following.**
>
> > I think the scientific statement is quite clear here and the paper is worth accepting; the only shame is that the authors did not apply this approach to a richer task than bAbI.
>
> We appreciate your invaluable feedback. The reason we did not apply our model to larger and complex tasks is, in such tasks, even with improvement, it is hard to clearly show whether the relation reasoning performance of the model contributed to the enhancement of task performance. To show the effect on the relational reasoning performance solely based on memory network, we choose to adopt bAbI task, $N^{th}$ farthest task, similar to other relational memory network researches. However, as you recommended, the experimental result on a larger and more complex task will be included in the final version of the paper.
>
> > Also it would have been nice to compare the approach to a multi-head attention transformer since these also use distributed representations (across heads).
>
> In the experiment section of our paper, we show the comparison results of other memory network models (RMC [1], STM [2]) which are adopting transformer like multi-head self-attention or itself is a generalized transformer model (UT [3]). We updated the details of such works in our paper. More specifically, RMC [1] is adopting multi-head attention for the relation searching process. And STM [2] (self-attentive associative memory) also internally extensively applies outer product type self-attention mechanism for relation finding, as its name represents. Furthermore, UT [3], Universal Transformer is an enhanced generalization of Transformer model for complex relational reasoning tasks.
>
> > The authors may be interested in the following architecture MERLIN which also uses a reconstruction loss to improve memory representations: https://arxiv.org/abs/1803.10760
>
> Thank you for recommending interesting research. It would be very helpful for our further research.
>
> [1] Santoro, Adam, et al. "Relational recurrent neural networks." Advances in neural information processing systems. 2018.
> [2] Le, Hung, Truyen Tran, and Svetha Venkatesh. "Self-Attentive Associative Memory." arXiv preprint arXiv:2002.03519 (2020).
> [3] Dehghani, Mostafa, et al. "Universal transformers." arXiv preprint arXiv:1807.03819 (2018).

---

### Official Review · AnonReviewer1 · 2020-10-29
**Multiple memories and reconstruction loss for MANNs**

**Rating:** 5
**Confidence:** 5

**Review:**

The paper proposes two extensions for existing memory networks (e.g., DNC) to improve associative reasoning: (a) multiple memory blocks instead of just one; and (b) self-supervised training as auxiliary tasks. Experiments conducted on toy datasets show that these extensions lead to improved performance over DNC and the method is somewhat competitive against newer variants on several synthetic tasks.

Pros:
- Understanding why MANN works and doesn't, and how to improve it are still open problems. And thus this work is welcomed.
- The empirical results are positive, suggesting the introduced ideas have merits.

Cons:
-  In terms of novelty, this work is quite incremental. The first extension is simply about dividing the external memory in DNC into multiple blocks that connect via an attentive gate. The argument of having multiple memory blocks is to diversify the representation of the same input. A similar idea has been introduced in [3], although the motivation was different. The second extension is a reconstruction loss on the input signals. To what degree it works as  “association reinforcing” as claimed is unclear, even though the loss would work as a regulariser as in standard hybrid loss in existing neural networks.
-  In terms of experiments, the tasks are too simple and particularly favor the model design in the paper. For example, in the copy task, we need faithful information from the input signals for correctly decoding (copying), which can be enhanced via the reconstruction loss during the encoding phase. The same thing applies to the associative recall task.
- Note that for both copy and associative recall, the length of the input sequence ([8, 32]) is much smaller than the number of memory slots (64) (see Appdx A.2.1). With such redundancy in storage, it is not very surprising to me that dividing the memory into smaller blocks can improve convergence.
- The main baseline used for comparison in this paper is DNC, which, in my opinion, is out-of-date as it can be outperformed easily (e.g., existing works on memory networks [1, 2] show significant improvements over DNC on these toy tasks).


Some minor comments:
- L_ar(i_t, y_t) is not defined in the paper. Do the authors use L1, L2 or binary cross-entropy loss?

[1] Relational recurrent neural networks, Santoro et. al., NIPS-2018
[2] Improving Differentiable Neural Computers through Memory Masking, De-allocation, and Link Distribution Sharpness Control, Csordas et. al., ICLR-2019
[3] Pham, Trang, Truyen Tran, and Svetha Venkatesh. "Relational dynamic memory networks." arXiv preprint arXiv:1808.04247 (2018).

---

> ### Author Response · Authors · 2020-11-18
> **Response to AnonReviewer1 (1/3)**
>
> **Thank you for your constructive and valuable comments. We’ve revised our paper following the suggestions and will explain your concerns in the following.**
>
> * Q1 (Contribution of our proposed methods)
>   * Our work is not a mere extension of DNC model. Our main contribution is showing that the ‘distributed representation’ concept [7, 8, 9, 10] can be applied to a content-based addressing memory model to obtain a more rich representation for relational reasoning from a given input sequence. In other words, we are proposing a novel way of retrieving relational information which can even replace the self-attention mechanism for MANN. Nowadays, most MANN models are focusing on using the attention or self-attention method to find relational information from the input sequence. However, we adopt a simple and unprecedented way of addressing the same problem. Without relying on a highly computational self-attention method, we still can obtain similar relational reasoning performance, even with DNC, and our parallel architecture and auxiliary task loss introduce negligible additional computational overhead compared to the attention mechanism. Furthermore, all these contributions are generally applicable to any MANN. As supporting evidence, we added the experimental result of another modified MANN model (RMC-AR, DMRMC-AR) that adopted our contributions to the Appendix of our paper. Therefore, our work is not just a simple incremental work of the usual MANN model, it provides a novel and efficient way for relational information finding method and also has generality as a policy which can be applied to any MANN model.
>   * Moreover, the idea introduced in [3], which you mentioned, is clearly different from our work. [3] is mainly purposed to store and retrieve structured data from memory. The structured data, such as graphs, already have a pre-defined relationship among its entities and there is no need to searching for the relational information between entities. The main point of [3] is how to store and retrieve such graph structure in the memory and use such structure for inference, not constructing the graph. Furthermore, the contents of multiple memory slots are not different representations of the same input data. They are disjoint parts of the input graph and it is nothing to do with distributed representation or distributed memory. Compared to [3], our DAM makes no assumption about the input data structure other than the sequential arrival. And, by itself, learns the relationship between input items and encodes it into the distributed representation in the memory.
>   * For the clarification of association performance of AR loss, it can be explained by multi-task learning theory. In a multi-task learning setting, well designed auxiliary task can allow the model to learn beneficial representations for the main task [2, 4, 5, 6]. In our case, ARL not only helps the main task of the model to learn better representation by reinforcing the relationships that exist in input sequence according to a given task but also enhances the memorization performance of the memory network by keeping refreshing input contents from the memory. The ‘association reinforcing’ effect comes from both ARL task’s rememorizing of representations in memory and its multi-task learning setting with the main task. In ARL’s own task, each item of the input sequence is sampled by Bernoulli trail of probability ‘p’. And whenever an input item is sampled, its loss is computed with the prediction output of the model at that time step. In this way, ARL is summed up and added to the target objective with a scaling factor. When the length of the input sequence is ‘n’, then the sequence of the Bernoulli trial becomes the binomial distribution with the probability of ‘p’ and the total number of trials ‘n’. Therefore, on average, the expected value of ARL sampling becomes ‘np’. This means ‘np’ portion of the input sequence is stochastically sampled and reconstructed by the model. This stochastic sampling method enables the mode to refresh ‘np’ amount of input sequence from the memory. Therefore, the reconstruction process enhances the overall memorization performance of the underlying memory network model. Similar approach with different purposed work in [1] provides clear evidence of ARL’s effect.

---

> > ### Author Response · Authors · 2020-11-18
> > **Response to AnonReviewer1 (2/3)**
> >
> > * Q2 (Experiments on Algorithmic task)
> >   * The experiments we adopted for our research are two kinds. First, the experiments for the verification of basic memory network functionalities (Copy, Associative Recall), and Second the experiments for the whole model performance on complex relational reasoning tasks (bAbI, $N^{th}$ farthest). The experiments you are mentioning are the basic memory functionality experiments (Copy, Associative Recall) and they are commonly used tasks for the evaluation of MANN. Your concern that ‘tasks are too simple’ can be addressed by the second type of experiments on complex relation reasoning tasks.
> >   * The purpose of first kind tasks is verifying the basic functionality of the given MANN. Algorithmic tasks (Copy, Associative Recall) are used to show the simple relation retrieving performance of memory networks and commonly adopted in papers such as NTM [11], SDNC [12], NUTM [13], DNC-MDS [14], etc. And Copy task is also adopted in many researches (NTM [11], DNC [15], NUTM [13], DNC-MDS [14], SDNC [12], STM [16], MNM-p [17], RIM [18], RMC [19]) to show the memorization performance of memory network, which is a crucial function of memory network. Since we integrated two main contributions, distributed memory architecture and ARL, to our baseline model, we need to verify its performance as a memory network model and show how much such modifications can improve each aspect of the memory network. Therefore, as an ‘Ablation study’, we used two tasks to show which aspect of the memory model is enhanced by which modification. After those basic experimental verifications as a memory network, we evaluated our model with complex tasks such as $N^{th}$ farthest task or bAbI task. For further verification, we also updated the manuscript with experimental results on Convex hull task. The experiments you mentioned are intended to show the basic functional performance verification of our model. And experimental results on more advanced tasks are also shown in the experiment section of our paper.
> >
> > [1] Trinh, Trieu H., et al. "Learning longer-term dependencies in rnns with auxiliary losses." arXiv preprint arXiv:1803.00144 (2018).
> > [2] Caruana, Rich, and Virginia R. De Sa. "Promoting poor features to supervisors: Some inputs work better as outputs." Advances in Neural Information Processing Systems. 1997.
> > [3] Pham, Trang, Truyen Tran, and Svetha Venkatesh. "Relational dynamic memory networks." arXiv preprint arXiv:1808.04247 (2018).
> > [4] Ben-David, Shai, and Reba Schuller. "Exploiting task relatedness for multiple task learning." Learning Theory and Kernel Machines. Springer, Berlin, Heidelberg, 2003. 567-580.
> > [5] Alonso, Héctor Martínez, and Barbara Plank. "When is multitask learning effective? Semantic sequence prediction under varying data conditions." arXiv preprint arXiv:1612.02251 (2016).
> > [6] Rei, Marek. "Semi-supervised multitask learning for sequence labeling." arXiv preprint arXiv:1704.07156 (2017).
> > [7] Fodor, Jerry A., and Zenon W. Pylyshyn. "Connectionism and cognitive architecture: A critical analysis." Cognition 28.1-2 (1988): 3-71.
> > [8] Chalmers, David J. "Syntactic transformations on distributed representations." Connectionist natural language processing. Springer, Dordrecht, 1992. 46-55.
> > [9] Ferrone, Lorenzo, and Fabio Massimo Zanzotto. "Symbolic, distributed and distributional representations for natural language processing in the era of deep learning: a survey." arXiv preprint arXiv:1702.00764 (2017).
> > [10] Le, Quoc, and Tomas Mikolov. "Distributed representations of sentences and documents." International conference on machine learning. 2014.
> > [11] Graves, Alex, Greg Wayne, and Ivo Danihelka. "Neural turing machines." arXiv preprint arXiv:1410.5401 (2014).
> > [12] Rae, Jack, et al. "Scaling memory-augmented neural networks with sparse reads and writes." Advances in Neural Information Processing Systems. 2016.
> > [13] Le, Hung, Truyen Tran, and Svetha Venkatesh. "Neural stored-program memory." arXiv preprint arXiv:1906.08862 (2019).
> > [14] Csordás, Róbert, and Juergen Schmidhuber. "Improving differentiable neural computers through memory masking, de-allocation, and link distribution sharpness control." arXiv preprint arXiv:1904.10278 (2019).
> > [15] Graves, Alex, et al. "Hybrid computing using a neural network with dynamic external memory." Nature 538.7626 (2016): 471-476.
> > [16] Le, Hung, Truyen Tran, and Svetha Venkatesh. "Self-Attentive Associative Memory." arXiv preprint arXiv:2002.03519 (2020).
> > [17] Munkhdalai, Tsendsuren, et al. "Metalearned neural memory." Advances in Neural Information Processing Systems. 2019.
> > [18] Goyal, Anirudh, et al. "Recurrent independent mechanisms." arXiv preprint arXiv:1909.10893 (2019).
> > [19] Santoro, Adam, et al. "Relational recurrent neural networks." Advances in neural information processing systems. 2018

---

> > > ### Author Response · Authors · 2020-11-18
> > > **Response to AnonReviewer1 (3/3)**
> > >
> > > * Q3 (Model's large memory slot size compared to input sequence size on Algorithmic task)
> > >   * As you pointed out, we reconfigured our model to use a smaller number of memory slots compared to input sequence length and re-evaluated our model performance. And updated the manuscript accordingly (In Appendix G.2). The revised experimental results more clearly showed the benefits of DAM-AR architecture. We appreciate your constructive comments for revising our experimental verifications.
> > >
> > > * Q4 (Baseline model)
> > >   * The major contribution of our paper is that complex relational information can be learned through distributed representations of input and ARL, instead of relying on computationally expensive ‘attention’ based approaches. For this purpose, we need a baseline model that does not majorly design to use self-attention or other operations for relational information searching. Although DNC looks like an out-of-date model, it is quite appropriate because it is not designed to extensively searching for relational information from input sequences, and it has a well-performing content-based addressing memory mechanism. If we apply our modifications to DNC and then it shows comparable relational reasoning performance to the state-of-the-art MANN, then it is clear proof that our hypothesis is correct. If we adopt other more recent MANN as a baseline, they all include their own way of using self-attention and relation searching operations. Such a model’s architectural consideration for relation is overlapping redundancy for our intention of proving the hypothesis, and even with improvement, it is not evident whether our modifications are the source of enhancement.
> > >   * Furthermore, in our experiments, our model outperformed all other DNC variants and even other types of MANNs (self-attention based models, meta-learning based models). Most models have state-of-the-art performance, but our model showed superior or comparable performance with such models. The purpose of our research is not to pick a very good MANN model and incrementally improve it. Our main goal is proposing a new promising way to retrieving relational information other than self-attention.
> > >
> > > * Q5 (Loss function for ARL)
> > >   * We updated to describe adopted loss functions of $L_{ar}$ in Appendix B. The loss function used for ARL is chosen based on the property of the task and input data type. For bAbI task and algorithmic tasks, we adopted cross-entropy loss, and for $N^{th}$ farthest task, $L2$ loss is used.

---

> ### Author Response · Authors · 2020-11-23
> **Have additional concerns?**
>
> Dear AnonReviewer 1,
>
> We believe that we have addressed your concerns and clarified some of your points and revised our paper accordingly. Could you please let us know if you have any additional concerns or questions? We would be happy to provide further revisions or experiments to address any remaining issues and would appreciate a response from you for an updated impression.

---

### Official Review · AnonReviewer5 · 2020-11-06

**Rating:** 5
**Confidence:** 3

**Review:**

The paper introduces a modification to Differentiable Neural Computer called Distributed Associative Memory (DAM) that comprises of 1) multiple independent memory blocks and 2) association reinforcement loss (ARS). Experimentally DAM improves upon DNC on multiple tasks and is showing comparable performance to some relation-aware architectures.

**Paper strengths**
* It is interesting to see that a relatively simple modification can bring a prominent performance boost. While it probably still requires further investigation, the fact that an architecture with factorized memory blocks can match performance of an explicitly relational architecture suggests that  this is indeed a step in the right direction and/or the relational benchmarks currently used in the community are too simple.

**Paper weaknesses**
* Clarity. ARL which is one of the two novel components of DAM is not described clearly. $l_{ar}$ is not formally defined anywhere and its textual description is rather vague. What exactly does "sampled input sequence" mean? Should it be called "subsampled" instead? Is it always valid to simply subsample individual input tokens?
* Since the improved performance presumably comes from the multiple individual memory blocks, it is important to understand how exactly information is factorized across them and how each of the blocks is used. One can argue that a wide enough representation can potentially learn the factorization scheme and mimic it using multiple reading/writing heads. To me the basic intuition behind that is provided in the Introduction is not enough.
* I appreciate that authors do compare memory capacity across different DNC variants, but then it is important to do so for all the baselines and ideally evaluate all the baselines with the same number of floating point numbers reserved for memory. Otherwise, the exact source of the improvements is not clear.
* Authors may want discuss the following paper [1] which describes a highly relevant model.

I am happy to revise my score if the points above are addressed by authors.

** References **
[1] Marblestone, Adam, Yan Wu, and Greg Wayne. "Product Kanerva Machines: Factorized Bayesian Memory." arXiv preprint arXiv:2002.02385 (2020).

---

> ### Author Response · Authors · 2020-11-18
> **Response to AnonReviewer5 (1/3)**
>
> **Thank you for the constructive comments. We appreciate the comments for improving the clarity of statements and experimental verifications. The manuscript is revised accordingly, and the responses to your main concerns are listed below.**
>
> * Q1 (Clarity of ARL)
>   * ARL is an auxiliary task which reconstructs sampled input sequence based on memory contents in a multi-task learning setting. Its reconstruction task enhances association performance (relation finding) of the main task based on the multi-task learning scheme. The goal of an auxiliary task in MTL theory is to enable the model to learn representations that are shared or helpful for the main task [1, 2, 3, 4]. Similarly, ARL does this implicitly: it allows the model to learn beneficial representations for the main task and, at the same time, enhance memorization performance of a memory network.
>   * We added more descriptions of ARL to the manuscript for clarification. The formal definition of $L_{ar}$ is based on the error measure for estimating the difference between ‘reconstructed input’ and ‘original input’. And it is defined according to the task property of the input data type. We described it in the Appendix. For bAbI task, we used cross-entropy loss and for $N^{th}$ farthest task adopted $L2$ loss.
>   * In ARL sampling details, each input data item of the sequence is decided to be sampled by the Bernoulli trial of probability ‘p’. Therefore, when an input sequence is provided to the model, for each data item, the model decides whether it should sample an item or not. And whenever the input item is sampled, its loss is computed with the prediction output of the model at that time step. In this way, ARL is summed up and added to the target objective with a scaling factor. When the length of the input sequence is ‘n’, then the sequence of the Bernoulli trial becomes the binomial distribution with the probability of ‘p’ and the number of trials ‘n’. Therefore, on average, the expected value of ARL sampling becomes ‘np’. This means ‘np’ portion of the input sequence is stochastically sampled and reconstructed by the model. This stochastic sampling method enables the model to refresh ‘np’ amount of input sequence from the memory. This whole reconstruction task is trained with the main task in Multi-task Learning (MTL) setting.
>
> [1] Caruana, Rich, and Virginia R. De Sa. "Promoting poor features to supervisors: Some inputs work better as outputs." Advances in Neural Information Processing Systems. 1997.
> [2] Ben-David, Shai, and Reba Schuller. "Exploiting task relatedness for multiple task learning." Learning Theory and Kernel Machines. Springer, Berlin, Heidelberg, 2003. 567-580.
> [3] Alonso, Héctor Martínez, and Barbara Plank. "When is multitask learning effective? Semantic sequence prediction under varying data conditions." arXiv preprint arXiv:1612.02251 (2016).
> [4] Rei, Marek. "Semi-supervised multitask learning for sequence labeling." arXiv preprint arXiv:1704.07156 (2017).

---

> > ### Author Response · Authors · 2020-11-18
> > **Response to AnonReviewer5 (2/3)**
> >
> > * Q2 (Intuition about DAM architecture)
> >   * The basic intuition underlying DAM architecture can be explained in two ways. First, it is not a simple division of single large memory space, rather, it is a collection of separate independent memory spaces. In other words, each memory block has its own complete representation space. Therefore, the same input can be stored in many different encoded versions, similar to the multi-head attention in Transformer. Each version of input data can represent one of the diverse contexts or relations it represents in the sequence, the same as multi-head attention. And DAM learns to combine such diverse representation to a single rich one which can include more complex relational information, such as ‘multi-hop’ [5], in itself.
> >   * The second perspective is the ‘distributed representation’ concept for current input data. The distributed representation is a well-known concept which frequently adopted in NLP research literature. It is also conceptually adopted in a word embedding ‘Word2Vec’ or ‘Paragraph Vector’ in PV-DM [6]. As in such researches, a richer representation for a given task can be constructed by merging several other representations of input data. In other words, we can infuse task-related information with such a vector encoding procedure. In our case, several different representations of the same input data are distributed across multiple memory blocks and they are merged with attentional interpolation to produce ‘distributed representation’ of the input data.
> >   * Your concern that ‘wide representation (wide memory length) might play a similar role as multiple representations’ can be clearly explained by the scalability experiment of DAM. In the scalability experiment, we increase the length of a single representation in DNC (DAM-1) and compare its performance with the other cases which are adopting multiple smaller size of representations (DAM-2, 3, and 4). As shown in Fig.3 of our paper, for the same complex relational reasoning task (bAbI), the performance of the wide representation model (DAM-1) degrades as the size of the representation increases. In contrast, the multiple representation models (DAM-2, 3, and 4) show enhanced performance with an additional number of sub memory blocks. The intuition behind this improvement can be explained with a similar concept from the multi-head attention mechanism. In the content-based addressing memory, even though we increase the length of the representation, the representation subspace for encoding input data remains the same as before. Therefore, representation diversity does not improve at all. However, if we use multiple separate sub memory blocks, each independent memory block can have its own representation subspace. Such diverse representations of the same input data can provide more chance to the model to learn about sophisticated relational information that exists in the input sequence, such as ‘multi-hop’ [5] relations. For better understanding, we updated the manuscript with the visualized map of the attentive gate to show how the information is distributed across multiple memories.
> >
> > [5] Sukhbaatar, Sainbayar, Jason Weston, and Rob Fergus. "End-to-end memory networks." Advances in neural information processing systems. 2015.
> > [6] Le, Quoc, and Tomas Mikolov. "Distributed representations of sentences and documents." International conference on machine learning. 2014.

---

> > > ### Author Response · Authors · 2020-11-18
> > > **Response to AnonReviewer5 (3/3)**
> > >
> > > * Q3 (Scalability of DAM)
> > >   * The experiment you are mentioning is the scalability test for DAM. It is designed to show the performance difference between one large memory and several smaller memory blocks. However, the experimental setting you are suggesting is not correctly applicable, because, for each task, there is a minimum memory length for correct information storing in content-based addressing. During bAbI task, the reason we did not iteratively divide the fixed total memory size to have the collection of smaller memories is, it has a minimum representation length required for a sub memory block which does not cause information loss in bAbI task. In the external memory matrix augmented content-based addressing scheme, if the length of a memory slot becomes too small for a given task, the content of input cannot be encoded into storage correctly and the performance of content-based addressing memory severely degrades.
> > >   * We designed the scalability experiment to answer the following two questions, 1) “Which memory configuration is better for relational reasoning? ‘using a single large memory’ or ‘several numbers of smaller memory blocks’ when total memory size is same” and 2) “Can we get more performance enhancement if we use more memory blocks?”. To show the effect of distributed memory architecture compared to a single memory system, we designed the scalability experiment. The memory models we named DAM-2, 3, and 4 simply represent the number of sub memory blocks they used. In other words, the scalability experiment is showing the effect of varying the hyperparameter, “the number of sub memory blocks”, on relation reasoning task. It reveals the simply increasing the size of a single memory does not help the performance of the memory network. In Fig. 3 of the paper, the dotted line of orange color represents DNC model with the same total memory size for each of DAM-2, 3, and 4. As shown in Fig. 3, when the total memory size is the same, using several smaller sub-memory blocks is more helpful.
> > >   * However, as far as we understood your concern, we update our paper with a similar scalability experiment on Association Recall task which needs a smaller minimum memory block length and iteratively divides fixed size memory length until reaches the minimum length. The experimental result shows a similar performance pattern with the scalability test on bAbI. If we use more number of sub-memory blocks, the accuracy increases compared to the same size single large memory. If we fixed the total memory size and iteratively divide it into the smaller sub-blocks to construct DAM-2 and 3 before information loss occurs, then the results are also similar. Before the information loss occurs, the more blocks provide more performance.
> > >   * If we misunderstood your point or our answer does not fully address your concerns, please let us know. We are gladly responding with further explanation.
> > >
> > > * Q4 (Relevant model)
> > >   * Thank you for recommending an interesting paper. The paper you mentioned is a generative memory model which memorizes the distribution rather than an encoded representation of input data. It is a Bayesian memory and combining several smaller same memory networks with product factorization. The point that adopting multiple memories for the entire model is similar to ours, however, their work mainly focuses on memorizing good input distribution based on the Bayesian rule, and input data should be exchangeable episodes which means an order of input data does not matter. Therefore, it is hard to be applied to the sequential input data where ordering is important and has relational meaning. QnA task from NLP is one of such examples. In contrast, our model can effectively model relational information exist in such sequential input data according to the target task. Therefore, our model is more appropriate for NLP problems where input data is provided in order and it includes relational information. Furthermore, our model is more adaptive for the target task because our architecture can encode target task-related information while learning distributed representation of input data.

---

> ### Author Response · Authors · 2020-11-23
> **Have additional concerns?**
>
> Dear AnonReviewer 5,
>
> We believe that we have addressed your concerns and clarified some of your points and revised our paper accordingly. Could you please let us know if you have any additional concerns or questions? We would be happy to provide further revisions or experiments to address any remaining issues and would appreciate a response from you for an updated impression.

---

### Author Response · Authors · 2020-11-24
**Summary of revision**

We appreciate all reviewer's constructive comments and feedback. We updated our paper according to the reviewer's concerns as follows.

We,

* Updates $N^{th}$ Farthest result in Table 1.

* Add experimental results on Convex hull task (relational reasoning task) in Table 3.

* Add visualization of DAM's memory operation with respect to attentive gates in Appendix E.

* Describe the link between equations of Section 2 and equations of Section 3 [Response to Reviewer 3]

* Add descriptions of ARL for clarity in Section 3.2 [Response to Reviewer 1, 2, and 5]

* Add explanation about loss function used for ARL in Section 3.2 and Appendix B [Response to Reviewer 1 and 5]

* Add additional analysis on algorithmic tasks in Appendix G [Response to Reviewer 5]

* Add Transformer results for bAbI task [Response to Reviewer 2]

---

### Decision · Program_Chairs · 2021-01-07
**Final Decision**

**Decision:**

Reject

**Comment:**

After carefully reading the reviews and the rebuttal, and after going over the paper itself, I'm not sure the paper it ready for ICLR. I do believe there is a lot of useful content in the current manuscript, and I urge the authors to keep working on the manuscript and resubmit it in due time.

My concerns are as follows:
 (a) there is a lot of discussion about *relational information retrieval* -- however there is lack of any formalization of what this term means. I don't mind relational reasoning to be used as motivation, but when it is used to consider what are valid baselines and what are not, I feel compelled to understand what exactly it means. Why is *self-attention* retrieval not *relational*? Beside the task being seemingly relational in spirit, how do we test whether the retrieved mechanism carries any relational information whatsoever? I think the community had a learning lesson here in CLEVER dataset, which arguably does not require as much relational reasoning as it seemed. So I agree with Rev5, that there is a decent probability that the task we are using do not require relational information retrieval. While I understand that some of these systems are Transformer inspired, I feel transformer should be a baseline.
 (b) I also feel the paper should take one of two paths.
       - Either embrace larger scale tasks and baseline outside of the relational reasoning literature (like transformer) and particularly settings where potentially self attention will struggle due to the quadratic term or where they tend to be hard to train due to the difficulty of doing credit assignment through the attention mechanism
       - Provide more careful ablation studies and formalize the claims a bit more. Regarding e.g. the discussion of a single larger memory vs multiple memory blocks. One of the main difference comes from the attention over which memory block to use in the proposed approach, which due to softmax has a unimodal behavior. So is the reason why it works better this potential hiding of part of the memory representation (so a better way of reading a subset of the memory entry). This could potentially be done differently (e.g. multiplicative interaction in the same style, for e.g. that they were used in WaveNet). This is just a random thought on this particular aspect. I have similar questions about the self-supervised loss.

 I find the paper focusing on improving performance (unfortunately on toy domains) rather than ablation studies and an understanding and careful understanding of how things works. I realize there is some such analysis in the appendix. But I feel more of it should be in the main text. The paper is either proposing something that scales and works well at scale (and then understanding why is less important as it has direct application) or explores a very specific phenomena and then is fine to stay on toy tasks but there should be a bit of clarity in the claims, and an investigation whether the hypothesis (or intuition) put forward initially is the reason why the model works.